

**Revealing the Causes of Groundwater Level Dynamics in Seasonally Frozen Soil Zones**
**Using Interpretable Deep Learning Models**
Han Li [a,b,c], Hang Lyu [a,b,c]*, Boyuan Pang [a,b,c], Xiaosi Su [a,b,c], Weihong Dong [a,b,c], Yuyu Wan [a,b,c],
Tiejun Song [a,b,c], Xiaofang Shen [a,b,c]
[a]Key Laboratory of Groundwater Resources and Environment, Ministry of Education, Jilin
University, Changchun 130026, China
[b]Jilin Provincial Key Laboratory of Water Resources and Environment, Jilin University,
Changchun 130026, China
[c]Institute of Water Resources and Environment, Jilin University, Changchun,130021, China
*Corresponding author
Hang Lyu
E-mail: lvhangmail@163.com

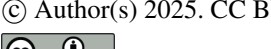

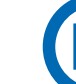

**Abstract**

14       Regional groundwater level prediction is crucial for water resource management,

especially in seasonally frozen areas. Accurate predicting groundwater levels during freeze–
thaw periods is essential for optimizing water resource allocation and preventing soil
salinization. Although deep learning models have been widely employed in groundwater level
prediction, they remain black boxes, making it difficult to simultaneously predict groundwater
levels and understand the dynamic causes. This study simulated the groundwater level
dynamics of 138 monitoring wells in the Songnen Plain, China, using a long short-term
memory (LSTM) neural network. The expected gradient (EG) method was applied to interpret
LSTM decision principles during different periods, revealing groundwater dynamics
mechanisms in seasonally frozen soil areas. The results showed that the LSTM model could
accurately simulate daily groundwater level trends, with 81.88% of monitoring sites achieving
NSE above 0.7 on the test set. The EG method revealed that atmospheric precipitation was the
primary source of groundwater recharge, while discharge occurred through evaporation, runoff,
and artificial extraction, forming three groundwater dynamics types: precipitation infiltration–
evaporation, precipitation infiltration–runoff, and extraction. During the freeze–thaw period,
groundwater levels in the precipitation infiltration–evaporation type decreased during the
freezing period and increased during the thawing period due to water potential gradient changes
driving soil–groundwater exchange. In contrast, the precipitation infiltration–runoff and
extraction types exhibited continuously increasing and decreasing trends, driven by recovery
after extraction and precipitation recharge. Our findings provide essential support for
groundwater resource assessment and ecological environmental protection in seasonally frozen
soil areas.
**Keywords:** Freezing-thawing process; Groundwater level dynamics; Seasonally frozen plain;
Interpretable deep learning models



## 1. Introduction

Groundwater levels are an external manifestation of the water balance within groundwater
systems, and changes in the groundwater level dynamics reflect the water budget of a region.
These changes also significantly impact regional ecological environments and the development
and utilization of water resources. A significant increase in groundwater levels may lead to
secondary soil salinization and marshland formation (Singh et al., 2012), while excessive
groundwater extraction may exacerbate water scarcity issues (Hao et al., 2014; Yang, 2012),
further causing geological problems such as land subsidence, ground collapse, and seawater
intrusion. Hence, gaining an in-depth understanding of the causes of groundwater level
dynamics and accurately simulating these changes can provide a theoretical basis for the
rational development and utilization of groundwater resources and for ecological environment
protection (Cai et al., 2022; Yating; et al., 2022).
Seasonally frozen soil areas are widely distributed globally. In China, they cover more
than half of the total land area, mainly in the northwest and northeast regions where water
scarcity is a prominent issue (Wang et al., 2019). Unlike non-frozen soils, seasonally frozen
soil is a unique water–soil system that contains ice, and changes in the ice content are
accompanied by the dynamic storage of liquid water and dynamic changes in heat (Wu et al.,
2023). The movement and storage behavior of groundwater in these regions differ from those
in warm, non-frozen areas (Ireson et al., 2013), as the freeze–thaw process results in more
frequent interactions between soil water and groundwater (Lyu et al., 2023; Lyu et al., 2022;
Miao et al., 2017; Daniel and Staricka, 2000). This leads to significant differences in the causes
of groundwater level dynamics between the freeze–thaw and non-freeze–thaw periods in
seasonally frozen soil areas, making it more challenging to accurately simulate the regional
groundwater levels.
Current models used for groundwater level simulations can be broadly divided into two





categories: physical models and machine learning models (Ao et al., 2021). Most physical
models are based on hydrodynamic mechanisms and the principle of water balance. However,
in regions with complex geological structures or groundwater level dynamics, the application
of physical models becomes challenging due to the lack of data describing the spatial
heterogeneity of aquifers and temporal changes in boundary conditions (Raghavendra. N and
Deka, 2014). Hence, there are few simulation studies on regional-scale groundwater level
dynamics in seasonally frozen soil areas. In comparison, machine learning models have
demonstrated significant advantages in simulating groundwater levels. These models explore
the nonlinear relationships between inputs (such as meteorological and topographic data) and
outputs (groundwater level) without the need to consider internal physical mechanisms (Rajaee
et al., 2019), nor do they require predefined parameters such as hydraulic characteristics or
boundary conditions (Ao et al., 2021). Despite this, machine learning models typically
outperform physical models in terms of simulation accuracy, particularly in medium-to-long-
term simulation studies (Rahman et al., 2020; Ebrahimi and Rajaee, 2017; Fienen et al., 2016;
Demissie et al., 2009). One of the most successful deep learning architectures for modeling
dynamic hydrological variables is the long short-term memory (LSTM) network (Jing et al.,
2023; Wu et al., 2021). The LSTM model, which is an improved version of the recurrent neural
network (RNN), can more effectively capture long-term dependencies in time-series data
(Hochreiter and Schmidhuber, 1997). In the seasonally frozen soil regions of Northwest China,
14 years of continuous groundwater level simulations have shown that the LSTM model can
effectively handle long-term data and accurately simulate groundwater levels in seasonally
frozen soil areas (Zhang et al., 2018).

Although numerous studies have demonstrated the accuracy and predictive power of data-

driven models in hydrological fields, these models are essentially black boxes and cannot
explicitly explain the underlying physical processes and mechanisms (Zhou and Zhang, 2023).





To address this limitation, researchers have proposed various methods to interpret deep learning
models. Two widely used methods in groundwater research are the expected gradient (EG)
method (Jiang et al., 2022) and the Shapley additive explanations (SHAP) algorithm (Lundberg
and Lee, 2018). The broad application of the SHAP method is mainly attributed to its ability to
reveal, from a local perspective, the contribution of each input variable to the corresponding
model output at each time step (Wang et al., 2022) and, from a global perspective, the overall
influence of input variables on the model output over the entire simulation period (Niu et al.,
2023; Liu et al., 2022). However, the limitation of the SHAP method is that its interpretation
of input factors is static and independent, making it ineffective in capturing the complex
interactions between groundwater levels and long-term recharge and discharge dynamics. In
contrast, the EG method (Jiang et al., 2022) calculates the EG values of the input variables over
a specified time range, allowing for a better quantification of the impact of dynamic input
variables on output variables at a particular time. This capability theoretically makes the EG
method advantageous in groundwater level simulations with dynamic characteristics,
particularly in explaining the temporal effects of meteorological changes on groundwater level
across different periods. Nevertheless, there are currently no dedicated studies on the use of the
EG method to explain the causes of groundwater level dynamics, and its effectiveness in
understanding the relatively complex mechanisms of groundwater level dynamics in seasonally
frozen soil areas requires further validation.
In this study, the seasonally frozen soil area of the Songnen Plain in Northeastern China
was taken as an example. Through an in-depth analysis of three years of continuous monitoring
data from phreatic wells in this region, combined with meteorological, hydrological, and soil
texture data, the LSTM model was used to simulate the groundwater level dynamics. The
reverse interpretation technique, i.e., the EG method, was applied to explore the decision
principles of the deep learning model in simulating water levels during the non-freeze–thaw



and freeze–thaw periods, thus revealing the mechanisms behind groundwater level dynamics
across different periods in seasonally frozen soil areas. The research findings can demonstrate
and extend the application of interpretable deep learning models in the groundwater field,
providing essential support for groundwater resource assessment and ecological environment
protection in seasonally frozen soil areas.
**2. Data and methodology**
Figure 1 shows the workflow of this study, including three main steps. First, the LSTM
model is used to establish a nonlinear relationship between meteorological factors, human
activities, and groundwater level depths (Fig. 1a). The daily air temperature, precipitation,
extraction volume, and snow depth were used as input variables to predict the groundwater
level depths. Subsequently, the EG method (Jiang et al., 2022) was applied to the trained LSTM
model to obtain the EG scores of the input factors at different time steps. The EG scores
quantify the influence of the meteorological inputs (air temperature, precipitation, and snow
depth) and human activities (extraction volume) on the groundwater level depths during the
simulation process (Fig. 1b). Finally, the causes of groundwater level dynamics during the non-
freeze–thaw and freeze–thaw periods in seasonally frozen soil areas were identified.



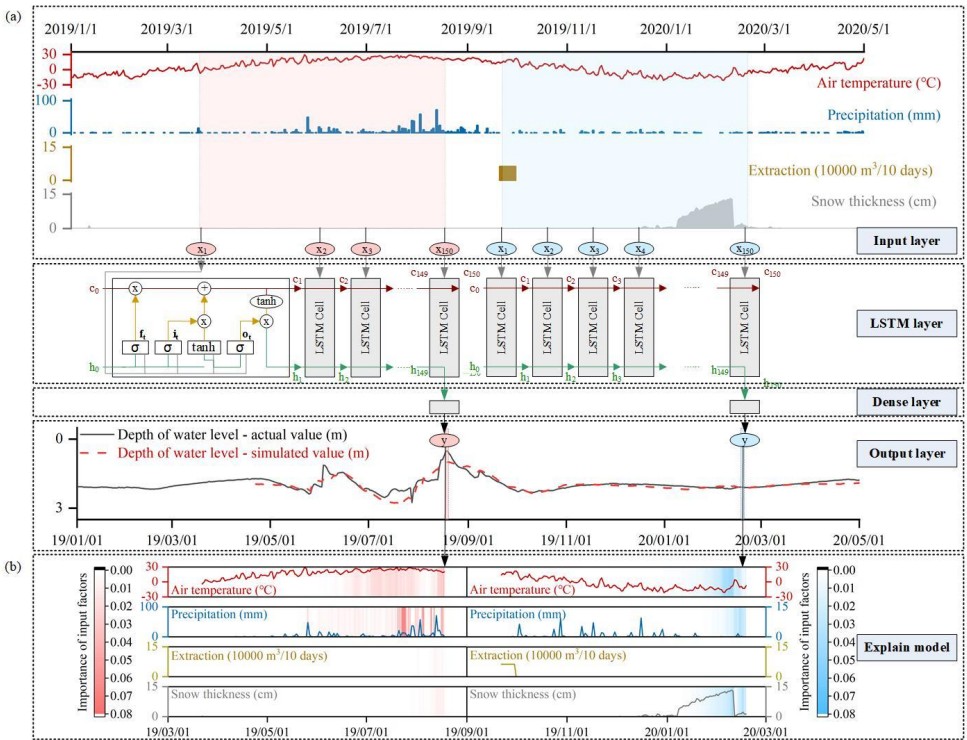


**Fig. 1.** Workflow of this study: (a) Model structure of the LSTM model, (b) EG scores of input

factors during the non-freeze–thaw and freeze–thaw periods.

### 2.1. Study area

The Songnen Plain is one of the three major plains in Northeast China. It is higher on the

periphery and lower at the center, with a total area of 182,800 km² (Fig. 2a). The study area is

surrounded by hills and mountains in the west, north, and east of the Greater and Lesser Xingan,

Zhangguangcai, and Changbai Mountains, respectively, and is connected to the West Liaohe

Plain by the micro-uplifted Songliao watershed in the south. The topography of the Songnen

Plain primarily comprises the eastern high plain, western piedmont sloping plain, western low

plain, and valley plain (Fig. 2b). The soil texture in the region mainly includes sandy loam,

sandy clay loam, clay loam, and loamy clay (Fig. 2c). The climate in the area can be mainly

characterized by two main types: first, it features a typical East Asian continental monsoon



climate with hot, rainy summers and cold, dry winters; second, although the distribution of the
climatic factors in the Songnen Plain is significantly influenced by latitude, there is a distinct
east–west difference, with arid conditions in the west and humid conditions in the east (Li et
al., 2022). The long-term average temperature of the Songnen Plain is 3.8 ℃, the long-term
average precipitation is 484.57 mm, and the long-term average evaporation is 1,498.1 mm. The
frost-free period ranges from 115 to 160 days. Freezing starts in mid-October from north to
south, and thawing begins in April from south to north. The freezing depth ranges from 1.5 to
2.4 m (Zhao et al., 2009) (Fig. 2d). The area is crisscrossed by rivers, with the Songhua River,
Nenjiang River, and their tributaries forming a centripetal drainage system. The lower reaches
of the Nenjiang River and Taoer River, as well as the Second Songhua River, flow through the
central plain from the north, west, and southeast, respectively.




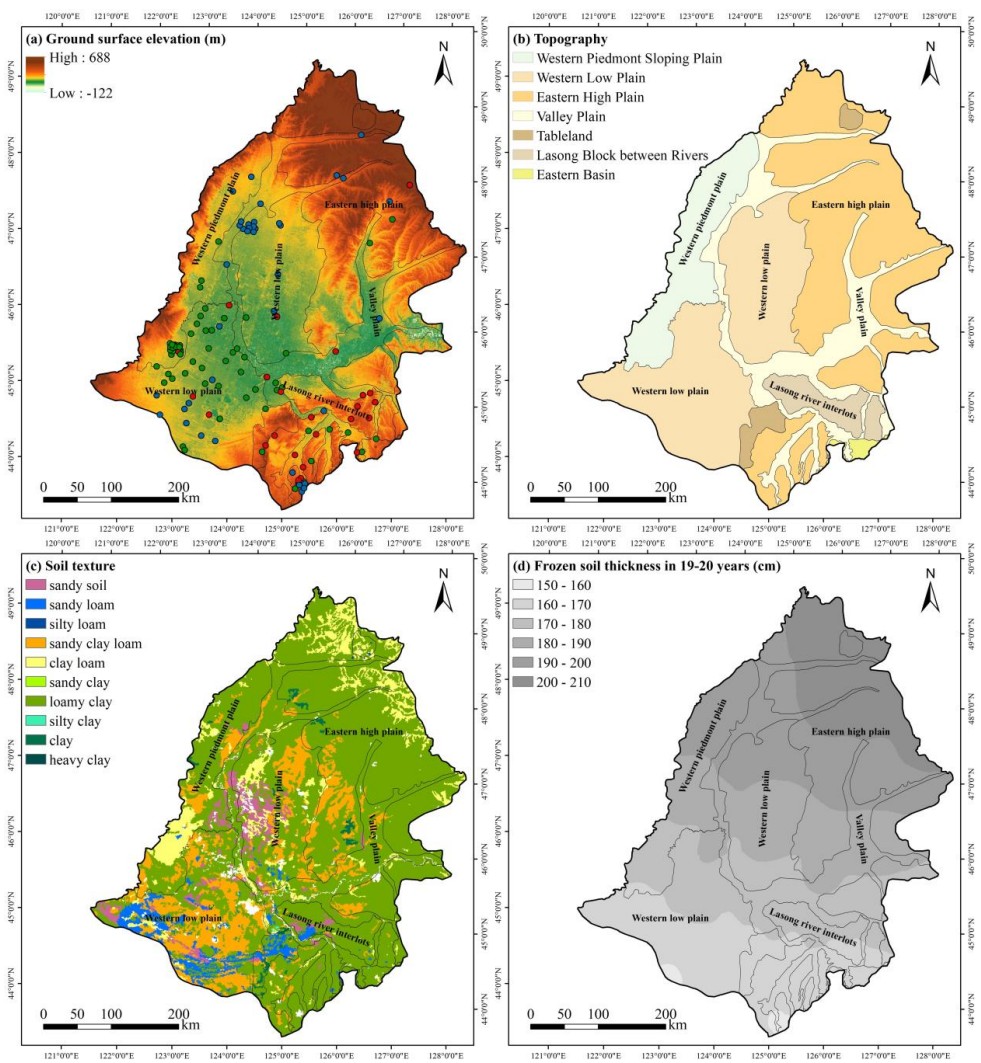


**Fig. 2.** Spatial distribution of the ground surface elevation (a), topography (b), soil texture (c)

and frozen soil thickness (d) in the Songnen Plain, China.

### *2.2. Dataset and selection of representative groundwater level values*

To simulate the dynamic changes in the groundwater level in seasonally frozen soil areas

and to analyze the driving mechanisms of groundwater level dynamics during freezing and

non-freezing periods, this study primarily used dynamic observational data from 2018 to 2021,

including precipitation, air temperature, snow depth, groundwater extraction volume, and



groundwater levels, as well as static data such as ground surface elevation and soil texture. The
precipitation and air temperature data were obtained from the "ERA5 hourly data on single
levels from 1979 to present" dataset, provided by the European Centre for Medium-Range
Weather Forecasts (ECMWF). ERA5 is the fifth-generation re-analysis of the global climate
and weather data with a spatial resolution of 0.25° × 0.25° and an hourly temporal resolution.
Daily snow depth data were sourced from the National Tibetan Plateau Data Center
(http://data.tpdc.ac.cn), with a spatial resolution of 25 km. The temporal and spatial resolution
of the groundwater extraction volume data was enhanced based on the spatial distribution and
water demand of major crops in the Songnen Plain, along with the precipitation data.
Groundwater level data from 138 phreatic wells were provided by the China Geological
Environment Monitoring Institute, while surface elevation data with a spatial resolution of 30
m were obtained from the Geospatial Data Cloud (https://www.gscloud.cn/search). Soil texture
data were sourced from the Resource and Environment Science and Data Center, compiled
from a 1:1,000,000 soil type map and soil profile data collected during the second national soil
survey of China.

To identify the causes of groundwater level dynamics during freezing and non-freezing

periods, representative groundwater levels were selected for analysis using the EG method at
different time periods. Based on the annual pattern of the groundwater level dynamics,
groundwater levels during the non-freezing period are influenced by human activities, flood-
season precipitation, and other factors, leading to greater fluctuations compared with that
observed in the freezing period. Therefore, selecting extreme values (either maximum or
minimum) as representative groundwater levels can effectively capture the peak or trough of
the groundwater level, reflecting the most significant state of groundwater recharge or
discharge during this period. Based on this, the trends in the groundwater level were analyzed
to identify the different dynamic characteristics during the non-freezing period. If the



groundwater level shows an overall uptrend, the maximum value represents the peak of the
recharge process; if it shows a downtrend, the minimum value reflects the maximum extent of
discharge.
However, during the freezing period, groundwater level fluctuations are relatively small,
and extreme values do not respond significantly to external factors. During this period,
groundwater levels may be influenced by soil freezing and thawing processes. Therefore, the
groundwater levels at critical moments of soil freezing and thawing were chosen as
representative values to more accurately reflect the response of groundwater level to
environmental changes. During the freezing period, after the "Beginning of Winter" solar term
(November 7–8), the average temperature continuously dropped to below 0 ℃, and a thin ice
layer gradually formed on the surface; after the "Rain Water" solar term (February 18–20),
temperatures increased, and the frozen soil began to thaw in both directions; finally, the frozen
soil fully thawed around the "Grain Rain" solar term (April 19–21) in spring (Lyu et al., 2023).
Therefore, the groundwater level at the "Rain Water" solar term was chosen as the
representative groundwater level during the freezing period to capture the rapid response of the
groundwater level to rising temperatures and thawing of the frozen soil.
*2.3. Research methods*
**2.3.1. LSTM model**
The LSTM neural network (Hochreiter and Schmidhuber, 1997) is an advanced RNN
widely applied in deep learning. It can store and associate previous information, effectively
addressing the issues of vanishing and exploding gradients that occur during the training of
long sequence data. The deep learning model used in this study comprises a single LSTM layer
and a dense layer. The LSTM layer is composed of recurrent cells arranged in a chain-like
structure, allowing information to be passed from the current time step to the next. The model
uses daily precipitation, air temperature, groundwater extraction volume, and snow depth from



the previous 150 days as input sequences to predict groundwater level depths. Each cell in the
LSTM layer includes four components: the input gate ($i_t$), the forget gate ($f_t$), the output gate
($o_t$), and the cell state ($c_t$) (as shown in the LSTM layer in Fig. 1a). The input gate determines
how much input information is transferred to the cell state. The forget gate primarily controls
how much information from the previous cell state is discarded and how much is carried
forward to the current moment. The output gate calculates the output based on the updated cell
state from the forget and input gates. The cell state is used to record the current input, the
previous cell state, and the information from the gate structures. In this study, we adopted the
LSTM equations proposed by Graves et al. (2013), which are represented by the following key
equations:
$$i_t = \sigma(W_{xi}x_t + W_{hi}h_{t-1} + b_t) \qquad (1)$$

$$f_t = f_t(W_{xf}x_t + W_{hf}h_{t-1} + b_f) \qquad (2)$$

$$c_t = f_t \odot c_{t-1} + i_t \odot tanh(W_{xc}x_t + W_{hc}h_{t-1} + b_c) \qquad (3)$$

$$o_t = \sigma(W_{xo}x_t + W_{ho}h_{t-1} + b_o) \qquad (4)$$

$$h_t = o_t \odot tanh(c_t) \qquad (5)$$

where the input and output vectors of the implicit layer of the LSTM at time step $t$ are $x_t$ and
$h_t$, respectively, the memory cell is $c_t$, and the values of the input, forget, and output gates are
$i_t$, $f_t$, and $o_t$, respectively. $W$ and $b$ represent the learnable weight and bias terms to be
estimated during the training period, respectively, σ(·) denotes the logistic sigmoid function,
tanh(·) is the hyperbolic tangent function, and $\odot$ represents elementwise multiplication.

Before training the model, the air temperature, precipitation, groundwater extraction

volume, and snow depth were normalized by mapping their values to a range between 0 and 1.
The adaptive moment estimation (Adam) algorithm (Kingma and Ba, 2014) was employed
during training, with an initial learning rate set to 0.03. The maximum training epoch number
was configured to 100, and an early stopping strategy was applied to prevent overfitting. For





each individual groundwater monitoring well, 70% of the input–output data pairs were
randomly sampled for training the LSTM model, and they were split into training and
validation samples at a ratio of 7:3. The training samples were repeatedly used to update the
model parameters until the loss function for the validation samples ceased to decrease. The
remaining 30% of the data were used for an independent evaluation of the model performance.
Random sampling allows for capturing the overall hydrometeorological variations observed
across different time periods.
**2.3.2. Model interpretations**

In 2017, Sundararajan et al. developed the integrated gradients (IG) method (Sundararajan

et al., 2017), which uses the gradient of the model's output to the input factors to infer the
specific contribution of the input variables to the output variable. The IG score for an input
factor $x$ (e.g., the precipitation at the i-th time step), representing the degree of contribution of
the input variable to the output variable, is expressed as follows:
$$\emptyset_i^{IG}(f, x, x') = (x_i - x_i') \int_{\alpha=0}^{1} \frac{\partial f\left(x' + \alpha(x - x')\right)}{\partial x_i} d\alpha \qquad (6)$$
where $\frac{\partial f\left(x' + \alpha(x - x')\right)}{\partial x_i}$ denotes the local gradient of the network $f$ at the interpolation point from
the baseline input ($x'$, when $\alpha = 0$) to the target input ($x$, when $\alpha = 1$).

However, the baseline input $x'$ in the above formula is a hyperparameter that must be

chosen carefully. In groundwater level studies, if the target input (e.g., a particular groundwater
level observation) is close to the chosen baseline input (e.g., long-term average groundwater
level), i.e., $x_i \approx x_i'$, the IG method may fail to capture the importance of current input factors,
such as precipitation or evaporation, on groundwater level changes (Sturmfels et al., 2020). To
address this issue, Jiang et al. (2022) developed the EG method, which is based on the IG
method but assumes that the baseline inputs follow the basic distribution D sampled from a
background dataset (such as the training dataset), thus avoiding the need to specify a fixed



baseline input. Given the baseline distribution D, the EG score $\emptyset_i^{EG}$ for the i-th input factor
can be calculated by integrating the gradients over all possible baseline inputs $x' \in D$, weighted
by the probability density function $p_D$. The EG score represents the influence of input factors
on the model output, with a higher absolute EG score indicating a greater impact of the
corresponding input factor on the model output, while an EG score close to zero suggests that
the input factor has little effect on the output. The EG score can be expressed as follows:
$$\emptyset_i^{EG}(f,x) = \int_{x'} \left( \emptyset_i^{IG}(f,x,x') \times p_D(x')dx' \right) \tag{7}$$
The above expression involves two integrals, which, according to Erion et al. (2021), can
both be considered expectations. Thus, the equation can be reformulated as:
$$\emptyset_i^{EG}(f,x) = E_{x'\sim D, \alpha\sim U(0,1)} \left[ (x_i - x_i') \int_{\alpha=0}^{1} \frac{\partial f\left(x'+\alpha(x-x')\right)}{\partial x_i'} \right] \tag{8}$$
**2.3.3. Evaluation metrics**
The evaluation metrics used in this study include the Nash–Sutcliffe efficiency (NSE)
coefficient and the root-mean-square error (RMSE). The NSE is used to assess the degree of
fit of the regression model. The RMSE quantifies how well the predicted values match the
observed values. If the NSE is close to 1 and the RMSE is close to 0, the model is more reliable.
$$NSE = 1 - \frac{\sum_{i=1}^{n}(x_i-y_i)^2}{\sum_{i=1}^{n}(x_i-\bar{x}_i)^2} \tag{9}$$
$$RMSE = \sqrt{\frac{\sum_{i=1}^{n}(x_i-y_i)^2}{n}} \tag{10}$$
where $x_i$ is the depth of the observed groundwater level, and $\bar{x}_i$ is the average value of $x_i$;
$y_i$ is the groundwater level depth simulated by the LSTM model; and *i* denotes the specific
sample ordinal number, from 1 to *n*.
**3. Results and discussion**
*3.1. Simulation Accuracy of Deep Learning Model for Groundwater Level*
A data-driven model (LSTM model) was used to simulate the daily groundwater level



depth of 138 aquifer monitoring wells in the Songnen Plain, China, from 2019 to 2021. Overall,
the simulation accuracy of the groundwater level depth was relatively high across the western
piedmont sloping plain, the eastern high plain, and the valley plain regions. In these areas, the
NSE values at the monitoring points in the test set ranged from 0.53 to 0.96 (Fig. 3a), with
87.14% of the monitoring points showing NSE values greater than 0.7. Over the entire
simulation period (including the training and test sets), the maximum error between the
simulated and observed values at each monitoring point mainly ranged from 0.5 to 2.5 m (Fig.
3b, d, and e), with 94.29% of the monitoring points having an average error of less than 0.5 m.
The annual groundwater level fluctuation at the monitoring points in this region was relatively
small, ranging from 0.41 to 6.54 m.

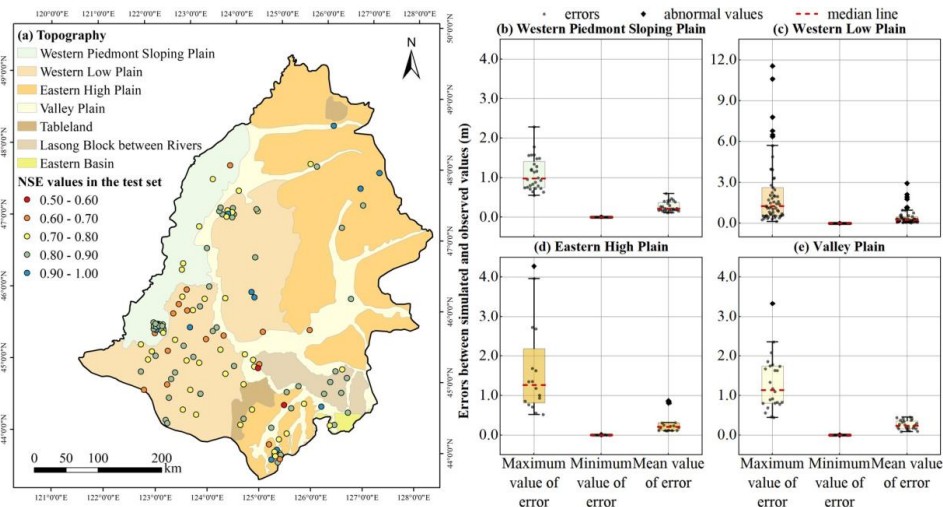


**Fig. 3.** (a) Spatial distribution of the NSE values on the test set for 138 groundwater level
monitoring points in the Songnen Plain, China. (b)–(e) Maximum, minimum, and mean errors
between simulated and observed groundwater levels at monitoring points in the western
piedmont sloping plain, western low plain, eastern high plain, and valley plain during the
simulation period.

Monitoring points with NSE values below 0.7 accounted for only 18.11% of the total





monitoring points in the study area, primarily located in the southern part of the western low
plain (Fig. 3a). In this area, the average error between the simulated and observed values of the
groundwater level depth at all the monitoring points ranged from 0.04 to 2.93 m; however, the
maximum error reached 11.56 m (Fig. 3c). The annual groundwater level fluctuation at the
monitoring points in this region was highly significant, with 21.43% of the monitoring points
exhibiting fluctuations greater than 10 m. This extreme variation in the groundwater level led
to a lower simulation accuracy of the LSTM model. In the data used to train the LSTM model,
the number of samples with extreme groundwater level depth values was relatively small, while
the number of samples with moderate values was higher, causing the model to be more inclined
to fit data within the moderate range of the groundwater level depth. Consequently, the
prediction accuracy of the model for extreme values was relatively low. Nonetheless, for
monitoring points in the southern part of the western low plain with higher simulation errors,
the simulated and observed groundwater level depth still exhibited similar dynamic
characteristics. The LSTM model could accurately capture the trends in the groundwater level
changes, with no significant lag between the simulated and observed values (Fig. 4).

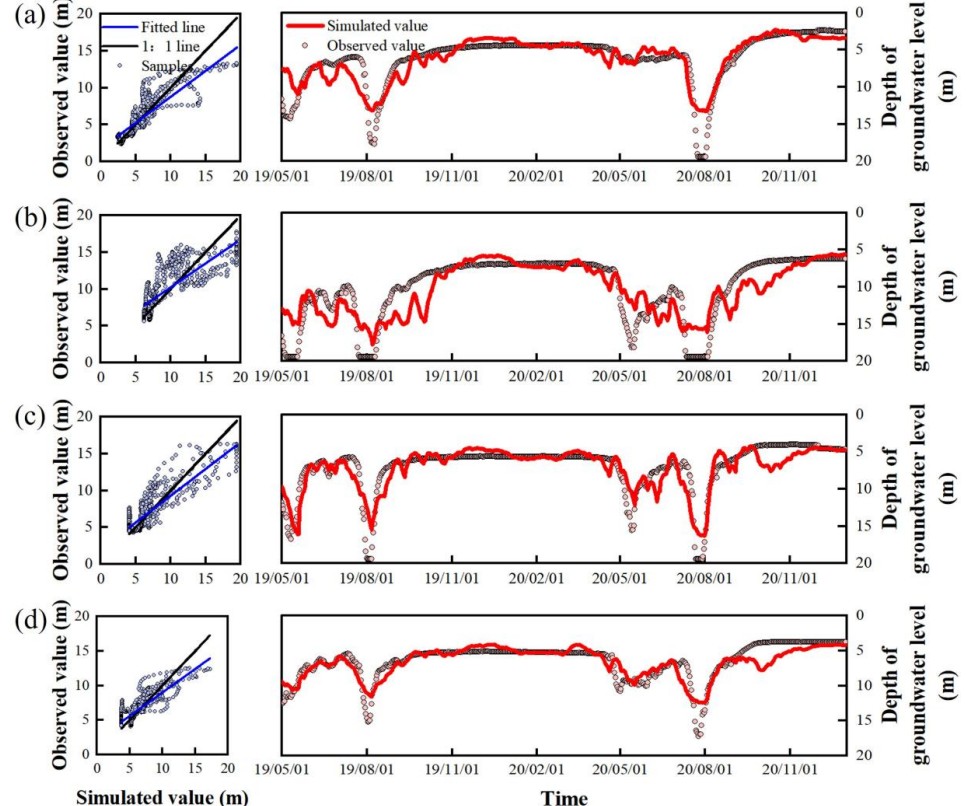

**Fig. 4.** Comparison of the simulated and observed groundwater level depths at typical points in the western low plain (NSE values on the test set < 0.7).

Overall, most of the groundwater monitoring points in the Songnen Plain, China, showed NSE values greater than 0.7 on the test set, indicating a relatively high simulation accuracy of the groundwater level depth based on the LSTM model. This suggests that the network structure of the LSTM model could accurately capture the dynamic relationships between the air temperature, precipitation, extraction volume, snow depth, and groundwater level.

***3.2. Dynamic Characteristics of Regional Groundwater Level and their Distribution Laws***

**3.2.1. Dynamic Characteristics of Annual Groundwater Level and their Spatial Distribution Laws**

Based on the characteristics of the annual groundwater level dynamic curves in the



Songnen Plain, China, the annual groundwater level dynamics can be categorized into three
types (Fig. 5).

The monitoring wells located in areas with a shallow groundwater level (less than 7 m) in

the northern part of the western low plain and valley plain (Fig. 5a) exhibited annual
groundwater level fluctuations of less than 4 m. Typically, the dynamic change in the
groundwater level is as follows: during the dry season from January to April, precipitation is
almost zero, and the groundwater level depth is significantly greater compared with those in
the other months; with the onset of the rainy season (May to August), precipitation increases,
causing the groundwater level to rise; after the rainy season ends (September to December),
the groundwater level depth gradually increases with decreasing precipitation (Fig. 5b). This
dynamic type of the groundwater level is the first annual dynamic type in the Songnen Plain,
with its corresponding monitoring wells accounting for 29.0% of all wells in the study area.

The monitoring wells located on Tableland, the Lasong Block between rivers, and the

eastern high plain (Fig. 5a) have relatively greater groundwater level depths, ranging from
approximately 5 to 11 m. From January to May each year, groundwater levels show a
continuous decline; with the increase in precipitation, the groundwater level begins to gradually
rise, reaching their annual peak in early October (Fig. 5c). The timing of the groundwater peak
is delayed by 1 to 2 months compared with the first dynamic type, indicating that the response
of the groundwater level to precipitation is slower (Fig. 5b and c). The annual groundwater
level fluctuation is within 5 m. This dynamic type is the second annual dynamic type in the
Songnen Plain, with its corresponding monitoring wells accounting for only 18.1% of all wells
in the study area.

In agricultural irrigation areas, such as the southern part of the western low plain and the

western piedmont sloping plain (Fig. 5a), the groundwater level depth typically ranges from 5
to 20 m. The dynamic curves of the groundwater level in the aquifer monitoring wells in these



areas exhibit distinct periodicity, showing a funnel-like and sawtooth pattern. The lowest

groundwater levels typically occur in May or August, while the highest level typically occurs

in November or later (Fig. 5d). During the irrigation season, groundwater levels drop

significantly, with annual fluctuations being generally within 15 m. This dynamic groundwater

type is widely distributed in the study area, with its corresponding monitoring wells accounting

for 52.9% of all wells, representing the third annual dynamic type in the Songnen Plain.

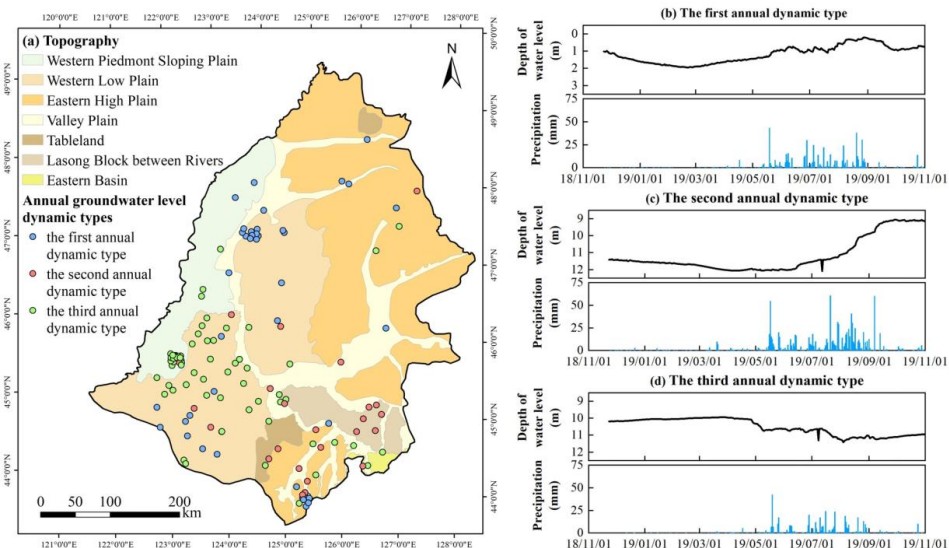

**Fig. 5.** (a) Spatial distribution of different annual groundwater level dynamic types in the

Songnen Plain, China; (b–d) Dynamic curves of different annual groundwater types and their

corresponding precipitation variations. (b) The first annual dynamic type is represented by an

unconfined aquifer monitoring well, numbered 230204210070, located in the western low plain;

(c) The second annual dynamic type is represented by an unconfined aquifer monitoring well,

numbered 220182210411, located in the Lasong Block between rivers; (d) The third annual

dynamic type is represented by an unconfined aquifer monitoring well, numbered

220802210145, located in the western piedmont sloping plain.



**3.2.2. Dynamic Characteristics of Groundwater Level During the Freeze–thaw Period and their Spatial Distribution Laws**

Freeze–thaw processes increase the frequency of interactions between soil water and groundwater (Lyu et al., 2022; Miao et al., 2017; Daniel and Staricka, 2000). As a typical seasonally frozen soil region, the Songnen Plain, China, exhibits three main forms of the dynamic curves of the groundwater level during the freeze–thaw period: "decline during freezing, rise during thawing," "continuous decline," and "continuous rise" (Fig. 6). The monitoring points of the different dynamic types during the freeze–thaw period accounted for 38.4% (V-shaped), 23.2% (continuous decline type) and 38.4% (continuous rise type), respectively.

At monitoring points with a "V-shaped" groundwater level dynamic curve, characterized by "decline during freezing, rise during thawing" (Fig. 6a), the groundwater level fluctuated by approximately 0.2–0.9 m during the freeze–thaw period. The time when the groundwater level reached its maximum depth roughly coincided with the time when the soil reached its maximum frozen thickness. These monitoring wells are primarily distributed in areas with a shallow groundwater level in the northern part of the western low plain and the valley plain, with a few located in the southern part of the western low plain. At the beginning of the freezing period, groundwater level depths at these wells were typically within 5 m (Fig. 6d).

For the continuous decline and continuous rise types, the dynamic curves of the groundwater level during the freeze–thaw period exhibited either a "continuous decline" or "continuous rise" (Fig. 6b and c), with the rate of change remaining consistent throughout both the freezing and thawing periods. Monitoring points with the continuous decline in the groundwater level were mainly distributed in areas with deeper groundwater levels, such as the eastern high plain and the Lasong Block between rivers, where the groundwater level depth ranged from 4.52 to 11.51 m at the start of the freezing period (Fig. 6d). In contrast, monitoring





wells with a continuous rise in the groundwater level during the freeze–thaw period were
mainly found in agricultural irrigation areas such as the southern part of the western low plain
and the western piedmont sloping plain, where the groundwater level depth at the beginning of
the freezing period ranged from 4.71 to 19.91 m (Fig. 6d).

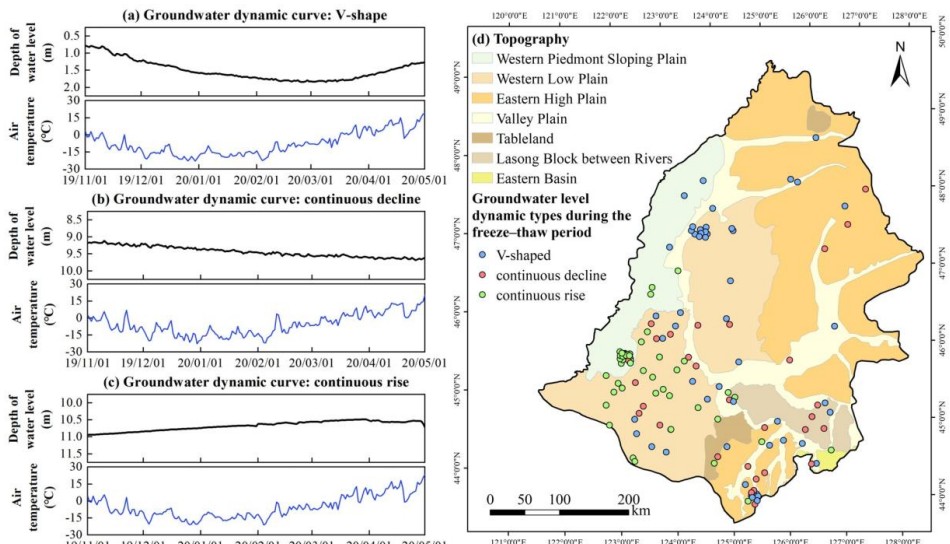


**Fig. 6.** (a–c) Dynamic curves of different groundwater types during the freeze–thaw period and
corresponding changes in air temperature; (d) Spatial distribution of different groundwater
level dynamic types during the freeze–thaw period in the Songnen Plain, China. The dynamic
curves of the groundwater level exhibiting patterns of (a) V-shaped, (b) continuous decline, and
(c) continuous rise correspond to the unconfined aquifer monitoring wells numbered
230204210070, 220182210411, and 220802210145, respectively.
***3.3. Main Controlling Factors and Identification of Causes for Various Groundwater Level***
***Dynamic Types***
After the application of the EG method to the trained models for the 138 groundwater
level simulations, the EG scores ($\phi_i^{EG}$) were obtained for precipitation, air temperature,
extraction volume, and snow depth within 150 days prior to the representative groundwater





level values for each annual and freeze–thaw period groundwater level dynamic type (Figs. 7
and 8).

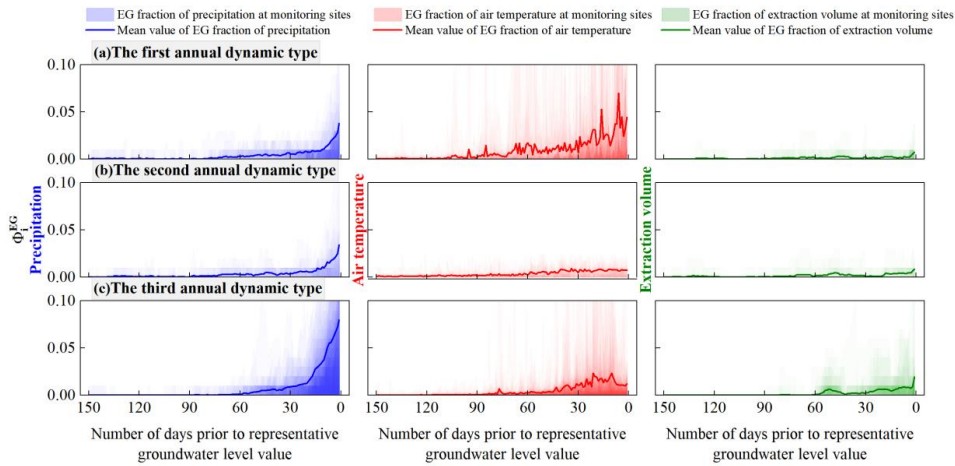


**Fig. 7.** EG scores ($\phi_i^{EG}$) of the precipitation, air temperature, and extraction volume for
different annual groundwater level dynamic types in the study area at different time steps.

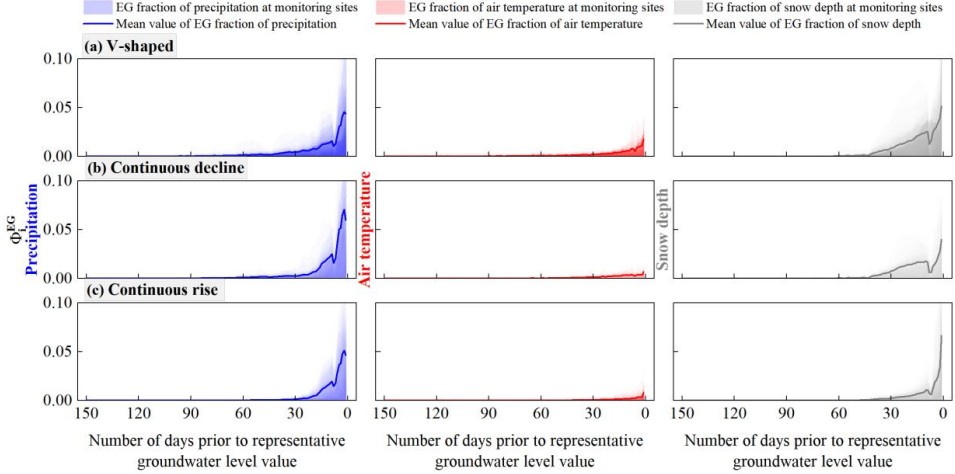


**Fig. 8.** EG scores ($\phi_i^{EG}$) of the precipitation, air temperature, and snow depth for different
groundwater level dynamic types during the freeze–thaw period in the study area at different
time steps.
**3.3.1. Main Controlling Factors and Identification of Causes for Annual Groundwater**



**Level Dynamic Types**


Within 90 days before the representative groundwater level values, the average EG scores
for the precipitation and air temperature in the first annual dynamic type ranged from 0 to 0.04
and from 0 to 0.07, respectively, while the average EG score for the extraction volume did not
exceed 0.01 (Fig. 7a). This indicates that the groundwater level depth in this dynamic type was
significantly influenced by precipitation and air temperature, while the effect of extraction was
negligible. Thus, the changes in the groundwater level depth may be related to the precipitation
infiltration–evaporation process. When a clear peak in precipitation occurred (Fig. 9b), the EG
score also increased significantly (exceeding 0.15), and the groundwater level rose accordingly
(Fig. 9e). Precipitation directly recharged the groundwater. Within the 90 days when
precipitation influenced the representative groundwater level value, a total precipitation of
408.09 mm led to an overall rise in the groundwater level by 1.12 m (Fig. 9b and e). During
periods without precipitation, the air temperature continued to rise (Fig. 9a), reflecting higher
soil evaporation. At this time, the EG score for the air temperature was also relatively high
(ranging from 0.10 to 0.20), and the groundwater level showed a slight decline (Fig. 9e). This
suggests that evaporation was the primary discharge mechanism for groundwater in this
dynamic type. Therefore, based on the groundwater recharge and discharge mechanisms, the
first annual groundwater dynamic type is summarized as the precipitation infiltration–
evaporation type.
In contrast, in the second annual dynamic type, only the precipitation had a significant
impact on the groundwater level depth within 90 days before the representative groundwater
level value (with the EG scores ranging from 0 to 0.03), while the average EG scores for the
air temperature and extraction volume remained between 0 and 0.01 (Fig. 7b). Precipitation
almost consistently recharged the groundwater during the 90 days before the representative
groundwater level values (with an average EG score of approximately 0.012), causing a gradual



rise in the groundwater level (Fig. 9j). However, the rate of groundwater rise was relatively
slow, with an average value of approximately 0.02 m/d. The air temperature fluctuated
significantly over the 90-day period (Fig. 9f), ranging from 4.41 to 28.57 ℃, but had no
significant impact on the groundwater level (Fig. 9j). The EG score during periods of high
temperatures was also below 0.01, indicating that evaporation had little effect on the
groundwater level. There was some groundwater extraction in local areas around July and
October (Fig. 9h); however, it had a minimal impact on the groundwater level, with the EG
scores remaining below 0.01. The relatively deep groundwater level (nearly 13 m) suggests
that this groundwater type was primarily discharged through runoff. Therefore, the second
annual groundwater dynamic type was classified as the precipitation infiltration–runoff type.

In the third annual dynamic type, the precipitation, air temperature, and extraction volume

had a significant impact on groundwater level within a shorter period before the representative
groundwater level values (within 60 days), with the average EG scores in the ranges of 0–0.08,
0–0.02, and 0–0.02, respectively (Fig. 7c). This dynamic type is mainly distributed in
agricultural irrigation areas, such as the southern part of the western low plain and the western
piedmont sloping plain (Fig. 5a). The main crops in these areas are rice, soybeans, and corn
(You et al., 2021), and their water demand is concentrated in the summer, particularly between
June and August (Zhenxiang; et al., 2022). During this period, the air temperature shows a
fluctuating uptrend (Fig. 9k), with the EG scores reaching a maximum of 0.02, indicating that
high temperatures increase soil evaporation and crop transpiration. This leads to a higher water
demand from the crops; however, the low rainfall was insufficient to meet this demand during
these periods (Fig. 9l, with a daily maximum precipitation of only 33.80 mm), necessitating
additional groundwater extraction for irrigation to maintain crop growth (Fig. 9m). As a result,
the EG score for the extraction volume reached approximately 0.20 during this period, and
groundwater level decreased accordingly (Fig. 9o). This dynamic type indicates that



groundwater recharge comes from precipitation infiltration, and groundwater extraction is the
main discharge mechanism. Thus, the third annual groundwater dynamic type was classified
as the extraction type.

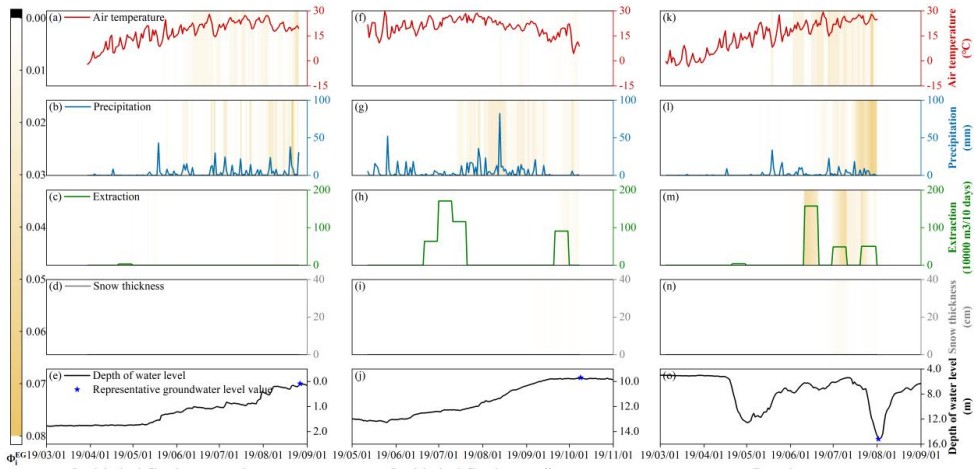

**Fig. 9.** Observed values and EG scores ($\phi_i^{EG}$) of the precipitation, air temperature, extraction
volume, and snow depth within 150 days before the representative groundwater level values
for various annual groundwater level dynamic types, as well as the corresponding annual
groundwater level depth dynamic curves. The precipitation infiltration–evaporation type,
precipitation infiltration–runoff type, and extraction type are represented by monitoring wells
230204210072, 220183210399, and 220821210024, with representative groundwater level
values corresponding to August 27, 2019, October 9, 2019, and August 2, 2019, respectively.
**3.3.2. Main Controlling Factors and Identification of Causes for Groundwater Dynamic**
**Level Types During the Freeze–Thaw Period**

A further analysis focused on the groundwater dynamic types during the freeze–thaw

period. In the V-shaped dynamic type, the average EG scores for precipitation and snow depth
within 60 days before the representative groundwater level values ranged from 0 to 0.05, while
the average EG score for the air temperature within 30 days before the representative
groundwater level values ranged from 0 to 0.02 (Fig. 8a). This suggests that the air temperature,



precipitation, and snow depth had a combined effect on the groundwater level depth of the V-shaped dynamic type during the freeze–thaw period. Within 30 days before the representative groundwater level values, the air temperature ranged from −21.10 °C to 4.40 °C, with the overall temperature being below 0 °C (Fig. 10b). As the air and soil temperatures dropped below 0 °C, the effective soil porosity decreased significantly due to water freezing, and the low-temperature suction related to the soil water potential between ice and water in the frozen soil increased gradually (Lyu et al., 2022). Under the combined effect of the capillary force and low-temperature suction, groundwater migrated upward continuously, thereby increasing the groundwater level depth (Fig. 10e). During this period, the snow depth increased with the decrease in temperature, reaching a maximum value of 13.22 cm on February 9, 2020 (Fig. 10d). The maximum EG score for the snow depth reached 0.03, indicating that snow had an impact on the groundwater level depth during the freeze–thaw period. When the air temperature exceeded 0 °C, the snow thawed rapidly (Fig. 10d), and the snow and frozen soil thaw water infiltrated to recharge the groundwater, causing the groundwater level to rise for the first time (Fig. 10e).

For the continuously declining and continuously rising dynamic types, only precipitation and snow depth affected the groundwater level depth during the freeze–thaw period. In the continuously declining groundwater dynamic type, the precipitation and snow depth influenced the groundwater level depth over a longer period before the representative groundwater level values (within 60 days), with the average EG scores below 0.07 and 0.04, respectively (Fig. 8b). In the continuously rising groundwater dynamic type, the average EG scores for the precipitation and snow depth within 30 days before the representative groundwater level values ranged from 0 to 0.05 and from 0 to 0.07, respectively, indicating that precipitation and snow depth affected the groundwater level depth in this dynamic type during the freeze–thaw period (Fig. 8c). Compared with precipitation and snow depth, the impact of the air temperature on





the groundwater level in both dynamic types was negligible (Fig. 8b and c), with the average
EG scores ranging from 0 to 0.01.

In both the freeze–thaw dynamic types, the air temperature fluctuated significantly over

the past 150 days (Fig. 10f and k), whereas the EG scores remained below 0.01, indicating that
the freeze–thaw effects had no significant impact on groundwater levels. Snow depth continued
to increase during the winter when the air temperature was below 0 °C (Fig. 10i and n). When
the air temperature rose above 0 °C, the snow gradually thawed, and the meltwater had some
recharging effect on groundwater levels (with maximum EG scores reaching 0.04). However,
due to the limited amount of snow and the high groundwater levels, the impact of snowmelt on
the groundwater level was gradual and limited, failing to significantly alter the original trends
in the continuously declining or continuously rising groundwater levels (Fig. 10j and o).
Therefore, the causes of the continuously declining and continuously rising groundwater level
dynamic types were related to the recovery process of the annual groundwater levels.

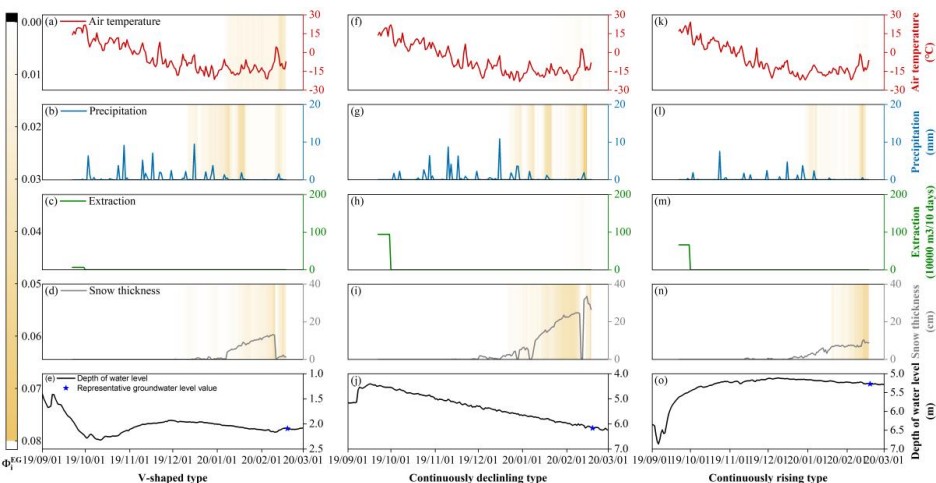


**Fig. 10.** Observed values and EG scores ($\phi_i^{EG}$) of the precipitation, air temperature, extraction
volume, and snow depth within 150 days before the representative groundwater level values
for various groundwater level dynamic types during the freeze–thaw period, as well as the
corresponding annual groundwater level depth dynamic curves. The V-shaped, continuous



decline, and continuous rise types are represented by monitoring wells 220106210371,
220182210410, and 220821210024, respectively. The representative groundwater level
corresponds to February 19, 2020.
***3.4. Regional Distribution Characteristics of the Dynamic Causes of Groundwater Level in***
***the Songnen Plain, China***

Based on the dynamic variations and spatial distribution characteristics of the groundwater

levels in the study area, groundwater monitoring points where the groundwater levels dropped
in the freezing period and rose in the thawing period, driven by soil freeze–thaw processes,
typically showed a precipitation infiltration-evaporation dynamic in terms of the groundwater
level dynamics during the year (Figs. 5b and 6a). These points were mainly distributed in areas
with shallow groundwater level depths, such as the northern part of the western low plain and
valley plain (Figs. 11a and 12a). Groundwater level dynamics unaffected by soil freeze–thaw
processes generally showed two trends: continuous decline or continuous rise (Fig. 6b and c).
Monitoring points with a continuous decline trend were mainly located in areas with a
significant groundwater level depth, such as the eastern high plain and the Lasong Block
between the rivers, where the annual groundwater level dynamics showed typical dynamic
characteristics of precipitation infiltration–runoff type (Fig. 5c). The monitoring points in
agricultural irrigation areas in the southern part of the western low plain and the western
piedmont sloping plain showed a continuous rise in the groundwater level during the freeze–
thaw period (Fig. 12a), and the dynamic type of the groundwater level in the year was mainly
the extraction type (Fig. 5d). Therefore, the "continuous decline" groundwater dynamic during
the freeze–thaw period was the recession phase of the groundwater level after the flood season
peak in the precipitation infiltration–runoff-type groundwater, while the "continuous rise"
groundwater dynamic was the recovery phase of the groundwater level after the extraction in
the extraction-type groundwater.



554 However, under the classification based on the freeze–thaw period, the proportions of the

555 V-shaped, continuous decline, and continuous rise types accounted for 38.4%, 23.2%, and 38.4%

556 of all monitoring points, respectively. These proportions did not completely align with the

557 annual classification of the precipitation infiltration–evaporation (29.0%), precipitation

558 infiltration–runoff (18.1%), and extraction (52.9%) types. This discrepancy can be partly

559 attributed to differences in the groundwater level depth. In some extraction monitoring points,

560 although the annual groundwater level dynamics showed typical extraction characteristics,

561 because the groundwater level at these monitoring points was shallow, the soil freezing and

562 thawing processes still had a significant impact on it, resulting in a V-shape water level change

563 at these points during the freeze–thaw period. The presence of such monitoring points increased

564 the proportion of the V-shape type during the freeze–thaw period, while reducing the proportion

565 of the continuous-rise type. Thus, the proportions of the freeze–thaw and annual classifications

566 were not entirely consistent, particularly in areas with a shallow groundwater level depth,

567 where soil freezing and thawing caused groundwater levels at some points of the extraction

568 type to exhibit V-shaped variations during the freeze–thaw period.

569 In the northern part of the western low plain, where groundwater level was shallow (less

570 than 5 m), the predominant annual groundwater dynamic was the precipitation infiltration-

571 evaporation type (Fig. 11a). Due to the proximity of the groundwater level to the surface, the

572 groundwater levels in these areas are more sensitive to meteorological factors. The dynamic

573 curves of the groundwater level show a characteristic in that the high water level period

574 corresponds to the rainy season. Specifically, in the Songnen Plain, peak precipitation and

575 groundwater level in this dynamic type occur simultaneously, typically between July and

576 August (Fig. 11d and f). The annual variation in the groundwater level was small, generally

577 less than 4 m (Fig. 11c). During the freeze–thaw period, the groundwater level dynamics in this

578 type exhibited a V-shaped pattern, with the groundwater level declining during the freezing



period and rising during the thawing period, with a fluctuation range of 0.2–0.9 m. However,
this V-shaped variation in the groundwater level is not accidental. At monitoring points with V-
shaped dynamics, the initial groundwater level depth and soil freezing depth at the beginning
of the freezing period were in the ranges of 0–5 m (Fig. 12d) and 1.6–2.1 m (Fig. 12c),
respectively. The soil was predominantly silty clay, with a maximum capillary rise height of up
to 5 m (Rui, 2004). Therefore, the initial groundwater level depth at these points was generally
less than the sum of the soil freezing depth and the maximum capillary rise height (Fig. 12a).
This means that during the freezing period, the low-temperature suction caused by soil freezing
and the pre-existing capillary forces in the soil form a complete hydraulic connection between
the frozen layer and the groundwater, causing the groundwater to continuously migrate toward
the freezing front during the freezing period.

Groundwater monitoring points exhibiting the precipitation infiltration-runoff type were

mainly distributed in the eastern high plain and the Lasong Block between rivers. In these areas,
the groundwater level is deeper, typically ranging from 5 to 12 m (Fig. 11b), and runoff is the
primary mode of groundwater discharge. The deeper groundwater level prolongs the infiltration
time of precipitation, resulting in a delayed response of the groundwater level dynamics to
precipitation recharge. Groundwater level peaks typically occur between August and October
(Fig. 11d), lagging behind the precipitation peak by approximately one month (Fig. 11f). Due
to the low recharge rate, groundwater level fluctuations are relatively moderate, with annual
variations generally within 4 m (Fig. 11c). During the freeze–thaw period, groundwater
monitoring points with continuously declining trends have greater initial groundwater level
depths, ranging from 4.52 to 11.51 m at the beginning of the freezing period (Fig. 12d). The
soil freezing depth in this dynamic type was shallower (between 1.6 and 1.8 m), and the soil
was still primarily silty clay (Fig. 12b and c). The greater groundwater level depth and
shallower soil freezing depth prevented a complete hydraulic connection between the frozen



soil and groundwater (Fig. 12a), resulting in the groundwater level being unaffected by the soil
freeze–thaw process.

In the agricultural irrigation areas of the southern part of the western low plain and the

western piedmont sloping plain, the groundwater level depth corresponding to the extraction
types typically ranged from 5 to 20 m (Fig. 11b). During the agricultural irrigation period,
significant groundwater extraction led to a marked decline in the groundwater level (Fig. 11c).
The low groundwater level period coincided with the peak extraction period, typically between
June and August (Fig. 11e and g). In areas with substantial groundwater extraction, a
groundwater depression cone had already formed, with annual groundwater level fluctuations
reaching up to 15 m (Fig. 11c). During the freeze–thaw period, the groundwater level dynamics
exhibited a continuous rising trend. In the southern part of the western low plain and the
western piedmont sloping plain, the initial groundwater level depth at the beginning of the
freezing period and the soil freezing depth were in the ranges of 5–20 m (Fig. 12d) and 1.6–
1.8 m (Fig. 12c), respectively, with the soil primarily comprising silty clay and sandy clay loam
(with a maximum capillary rise height of 3 m) (Fig. 12b). In this region, the initial groundwater
level depth was generally greater than the sum of the soil freezing depth and the maximum
capillary rise height, causing the hydraulic connection between the vadose and saturated zones
to be severed (Fig. 12a), and the groundwater level was unaffected by the soil freeze–thaw
process.





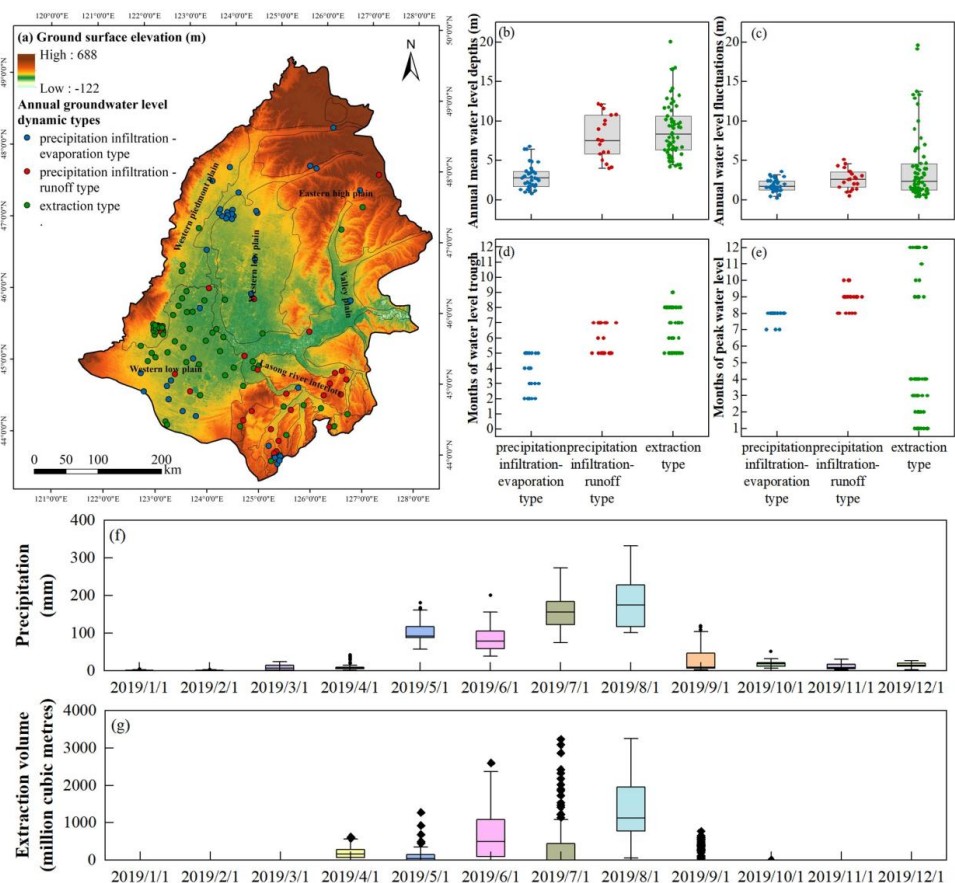

**Fig. 11.** (a) Spatial distribution of the ground surface elevation and three dynamic types of annual groundwater level (precipitation infiltration-evaporation type, precipitation infiltration-runoff type, and extraction type) in Songnen Plain, China. The correlation between the three dynamic types of annual groundwater level and (b) annual mean groundwater level depths, (c) annual water level fluctuations, (d) months of peak water level and (e) months of water level trough. (f) and (g) Monthly distribution of the precipitation and extraction volume in Songnen Plain, China in 2019, respectively. Each point in (b)–(e) represents a groundwater level monitoring point.






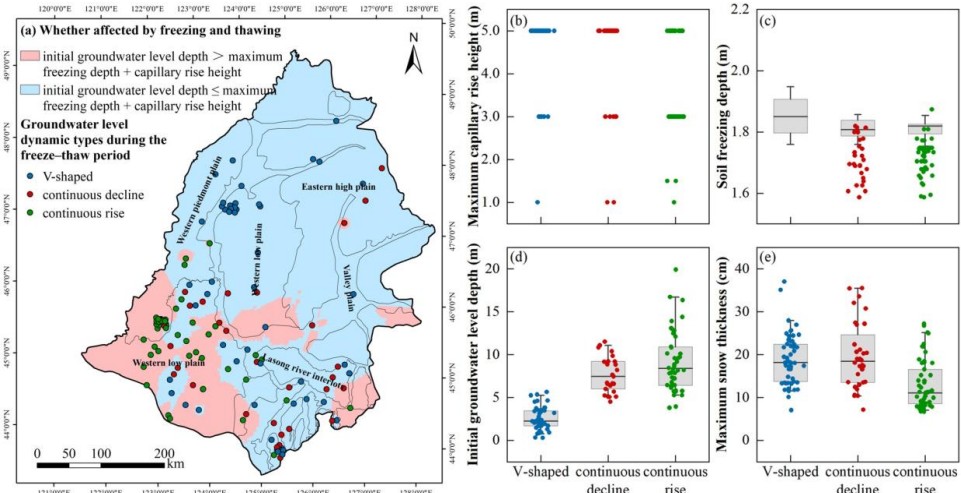

**Fig. 12.** (a) Spatial distribution of whether the groundwater level is affected by the soil freeze–
thaw process and the three groundwater level dynamic types during the freeze–thaw period (V-
shaped, continuously declining, and continuously rising) in the Songnen Plain, China.
Correlations between the groundwater level dynamic types in the three freeze–thaw period and
(b) maximum capillary rise height of the soil, (c) the soil freezing depth, (d) the initial
groundwater level depth at the start of the freezing period, and (e) maximum snow thickness.
Each point in (b)–(e) represents a groundwater monitoring well.
### *3.5. Limitations of existing models*
A deep learning model was successfully developed in this study to simulate the
groundwater level in the seasonally frozen ground regions of Northeast China, with 81.88% of
the monitoring wells in the study area achieving an NSE > 0.7 on the test set. A common issue
with deep learning models is that they are often considered black-box models, making it
difficult to interpret their internal decision-making processes, which limits their credibility and
interpretability in practical applications (Gunning; et al., 2019). In groundwater level
simulation studies, this research is the first to apply the EG method to quantify the importance
of input factors in simulating groundwater level during non-freezing and freezing periods,





revealing the driving forces behind groundwater level dynamics in different seasons. The
introduction of this method offers a novel approach to understanding the groundwater level
dynamics in seasonally frozen regions.

We opted for a local modeling approach (i.e., training a separate model for each

groundwater monitoring well) rather than a regional approach (training a single model with
data from multiple monitoring wells). This decision was based on our goal to identify the
contribution patterns of the input factors (precipitation, air temperature, extraction volume, and
snow depth) to groundwater level at the regional scale, including the duration of their influence
and the significance of their impact. From a prediction standpoint, a regional model might be
more suitable for areas where data are scarce or incomplete (Frame et al., 2022; Nearing et al.,
2021), as it can learn more general relationships between input and output factors from
historical data (Kratzert et al., 2019). However, regional models are associated with the issue
of multicollinearity between static factors, and this issue must be addressed. Collinear input
factors may share a substantial amount of information, making it difficult for the model to
accurately distinguish the independent influence of each input factor on the output, leading to
challenges in interpreting the impact of inputs on the output. Therefore, using regional models
to explain the causes of groundwater level dynamics in seasonally frozen regions could be more
challenging than using local models. Nevertheless, we acknowledge the advantages of regional
models. Future research could further explore how to address the multicollinearity issues
associated with static factors in regional models. In conclusion, we successfully combined deep
learning models with the EG method to reveal the causes of groundwater level dynamics in
seasonally frozen regions.
**4. Conclusions**

Through the application of interpretable deep learning methods, this study revealed the

causal mechanisms of the dynamic change in the groundwater level in seasonally frozen soil



areas. The groundwater level change characteristics at 138 monitoring sites were analyzed in-
depth through high-precision simulations based on the LSTM model. Combined with the
application of the EG method, this study elucidated the differences and causes of groundwater
level dynamics during the freeze–thaw period and throughout the year. The main findings of
the study are as follows:

First, the LSTM model demonstrated a high accuracy in simulating groundwater level

trends in seasonally frozen soil areas, with the NSE values on the test set ranging from 0.53 to
0.96. This indicated that the model could effectively capture the complex changes in the
groundwater level.

Second, through the application of the EG method, this study identified three main types

of groundwater level dynamics in the Songnen Plain of China throughout the year, namely,
precipitation infiltration–evaporation type, precipitation infiltration–runoff type, and extraction
type, accounting for 29.0%, 18.1%, and 52.9% of the total, respectively. During the freeze–
thaw period, these types manifested as "V-shaped," continuously declining, and continuously
rising trends, accounting for 38.4%, 23.2%, and 38.4%, respectively.

The recharge sources for all three types of annual groundwater level dynamics originated

from precipitation infiltration, while the discharge pathways were evaporation, runoff, and
artificial extraction, respectively. During the freeze–thaw period, a "V-shaped" groundwater
level trend indicated a significant influence of the soil freeze–thaw process on the groundwater
level. In contrast, the continuously declining and continuously rising trends reflected the
recovery processes following groundwater extraction and precipitation recharge, both of which
were unaffected by freeze–thaw processes. These dynamic types reflected the change patterns
of the groundwater level driven by multiple factors at different time scales.

The dynamic changes in the groundwater level in seasonally frozen soil areas were a

complex process influenced by various factors, such as aquifer properties, boundary conditions,



and freeze–thaw processes. In this context, deep learning models required more sophisticated
model structures and more input variables to more accurately capture the patterns in water level
fluctuations. To better understand the predictive mechanisms and feature importance within
these models, interpretative techniques, such as the EG method, are particularly valuable. In
this study, the EG method was applied to reveal the causal mechanisms behind groundwater
level dynamics, and future research may explore other advanced interpretability techniques to
further enhance our understanding of the model results. We believe that with the continuous
advancement of artificial intelligence, these methods will reveal even more valuable
information hidden within deep learning models. Moreover, the simulation accuracy of deep
learning models should not be the sole research focus, but rather a prerequisite. The true
significance of deep learning in the field lies in helping researchers gain deeper insights into
hydrological processes and ultimately discovering previously unknown laws and patterns.
Overall, this study demonstrated the significant potential of the EG method in explaining
the causal mechanisms behind groundwater level dynamics in seasonally frozen soil areas. This
approach not only resolves the conflict between model accuracy and interpretability but also
provides a new perspective for hydrological research. With the future introduction and
optimization of more interpretability methods, the EG method is expected to further reveal the
complexities of hydrological processes in seasonally frozen regions, providing more scientific
evidence for groundwater resource management and environmental protection.
**Credit authorship contribution statement**
Han Li: Conceptualization, Investigation, Formal analysis, Data curation, Visualization,
Writing–original draft. Hang Lyu: Conceptualization, Validation, Formal analysis, Resources,
Investigation, Data curation, Visualization, Supervision. Boyuan Pang: Investigation,
Visualization. Xiaosi Su: Investigation, Supervision. Weihong Dong: Resources, Data curation.
Yuyu Wan: Resources, Data curation. Tiejun Song: Data curation. Xiaofang Shen: Data



curation.
**Declaration of interests**
The authors declare that they have no known competing financial interests or personal
relationships that could have appeared to influence the work reported in this paper.
**Acknowledgments**
This research was supported by the Jilin Provincial Science and Technology Development
Plan Project (No.20230508036RC) and National Natural Science Foundation of China
(42172267, 42230204, U19A20107). Thanks to Groundwater Monitoring Project of China
Institute of Geo-Environment Monitoring for providing data support.

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



**Figure captions**
**Fig. 1.** Workflow of this study: (a) Model structure of the LSTM model, (b) EG scores of input
factors during the non-freeze–thaw and freeze–thaw periods.
**Fig. 2.** Spatial distribution of the ground surface elevation (a), topography (b), soil texture (c)
and frozen soil thickness (d) in the Songnen Plain, China.
**Fig. 3.** (a) Spatial distribution of the NSE values on the test set for 138 groundwater level
monitoring points in the Songnen Plain, China. (b)–(e) Maximum, minimum, and mean errors
between simulated and observed groundwater levels at monitoring points in the western
piedmont sloping plain, western low plain, eastern high plain, and valley plain during the
simulation period.
**Fig. 4.** Comparison of the simulated and observed groundwater level depths at typical points
in the western low plain (NSE values on the test set < 0.7).
**Fig. 5.** (a) Spatial distribution of different annual groundwater level dynamic types in the
Songnen Plain, China; (b–d) Dynamic curves of different annual groundwater types and their
corresponding precipitation variations. (b) The first annual dynamic type is represented by an
unconfined aquifer monitoring well, numbered 230204210070, located in the western low plain;
(c) The second annual dynamic type is represented by an unconfined aquifer monitoring well,
numbered 220182210411, located in the Lasong Block between rivers; (d) The third annual
dynamic type is represented by an unconfined aquifer monitoring well, numbered
220802210145, located in the western piedmont sloping plain.
**Fig. 6.** (a–c) Dynamic curves of different groundwater types during the freeze–thaw period and
corresponding changes in air temperature; (d) Spatial distribution of different groundwater
level dynamic types during the freeze–thaw period in the Songnen Plain, China. The dynamic
curves of the groundwater level exhibiting patterns of (a) V-shaped, (b) continuous decline, and
(c) continuous rise correspond to the unconfined aquifer monitoring wells numbered





230204210070, 220182210411, and 220802210145, respectively.
**Fig. 7.** EG scores ($\phi_i^{EG}$) of the precipitation, air temperature, and extraction volume for
different annual groundwater level dynamic types in the study area at different time steps.
**Fig. 8.** EG scores ($\phi_i^{EG}$) of the precipitation, air temperature, and snow depth for different
groundwater level dynamic types during the freeze–thaw period in the study area at different
time steps.
**Fig. 9.** Observed values and EG scores ($\phi_i^{EG}$) of the precipitation, air temperature, extraction
volume, and snow depth within 150 days before the representative groundwater level values
for various annual groundwater level dynamic types, as well as the corresponding annual
groundwater level depth dynamic curves. The precipitation infiltration–evaporation type,
precipitation infiltration–runoff type, and extraction type are represented by monitoring wells
230204210072, 220183210399, and 220821210024, with representative groundwater level
values corresponding to August 27, 2019, October 9, 2019, and August 2, 2019, respectively.
**Fig. 10.** Observed values and EG scores ($\phi_i^{EG}$) of the precipitation, air temperature, extraction
volume, and snow depth within 150 days before the representative groundwater level values
for various groundwater level dynamic types during the freeze–thaw period, as well as the
corresponding annual groundwater level depth dynamic curves. The V-shaped, continuous
decline, and continuous rise types are represented by monitoring wells 220106210371,
220182210410, and 220821210024, respectively. The representative groundwater level
corresponds to February 19, 2020.
**Fig. 11.** (a) Spatial distribution of the ground surface elevation and three dynamic types of
annual groundwater level (precipitation infiltration-evaporation type, precipitation infiltration-
runoff type, and extraction type) in Songnen Plain, China. The correlation between the three
dynamic types of annual groundwater level and (b) annual mean groundwater level depths, (c)
annual water level fluctuations, (d) months of peak water level and (e) months of water level



trough. (f) and (g) Monthly distribution of the precipitation and extraction volume in Songnen
Plain, China in 2019, respectively. Each point in (b)–(e) represents a groundwater level
monitoring point.
**Fig. 12.** (a) Spatial distribution of whether the groundwater level is affected by the soil freeze–
thaw process and the three groundwater level dynamic types during the freeze–thaw period (V-
shaped, continuously declining, and continuously rising) in the Songnen Plain, China.
Correlations between the groundwater level dynamic types in the three freeze–thaw period and
(b) maximum capillary rise height of the soil, (c) the soil freezing depth, (d) the initial
groundwater level depth at the start of the freezing period, and (e) maximum snow thickness.
Each point in (b)–(e) represents a groundwater monitoring well.