# Peer review of "Revealing the Causes of Groundwater Level Dynamics in Seasonally Frozen Soil Zones"

_EGUsphere, 2025_

## Author Comment (AC1)

This manuscript applies a machine learning (ML) approach to predict time-varying groundwater levels in seasonally freezing regions of China. The topic is timely and of high importance for groundwater resource management and environmental protection. However, the study overlooks several critical factors that could significantly influence the results and interpretations. By incorporating additional hydrogeological and environmental variables, the model's accuracy could be greatly improved, leading to a more comprehensive understanding of groundwater dynamics.

Response: Thank you for your valuable comments on our study. We have carefully reviewed your suggestions and made corresponding revisions, and we hope these modifications meet your expectations. We agree that key factors such as hydrogeological conditions and environmental variables may significantly influence the model outputs and their interpretation. However, the core focus of this study is on building LSTM models for each monitoring site individually, aiming to simulate the temporal variation of groundwater level at the point scale. Within this framework, spatially fixed attributes such as aquifer properties and topography remain relatively stable over the short term and are unlikely to exert dynamic influence on the time series at a single site. Additionally, factors such as vertical leakage and surface water interactions are difficult to quantify due to limited data availability. In future work, if data conditions permit, we will consider incorporating these variables to enhance the physical interpretability and predictive accuracy of the model.

Specific comments:

Line 25: Please define NSE upon first mention to ensure clarity for readers unfamiliar with the metric.

Response: Thank you for pointing out this issue. We have added the full term "Nash-Sutcliffe Efficiency" when "NSE" first appears in the abstract.

Line 39: Provide more detailed justification of why monitoring groundwater levels is crucial, not only for managing water resources but also for protecting

ecological systems. Additionally, consider using the ML-predicted results to present a case study with quantitative analysis to better illustrate the implications.

Response: Thank you for the valuable comments from the reviewer. Following the suggestions, we have comprehensively revised the relevant parts of the manuscript to further elaborate on the importance of groundwater level depth, especially emphasizing its role in water resource management and ecosystem protection. Additionally, we supplemented the citations with the study by Liu et al. (2022), which used machine learning to predict groundwater level depth in the lower Tarim River, providing a quantitative case validation of the practical significance of groundwater level prediction. The revised content is as follows:

"Groundwater level is a crucial indicator reflecting the water balance status of groundwater systems, and its dynamic changes reveal the evolving trends of regional hydrological processes. In terms of water resource management, monitoring groundwater level depth helps managers understand changes in groundwater storage, optimize water extraction schemes, and prevent resource depletion caused by overexploitation (Hao et al., 2014; Yang, 2012). Regarding ecosystem protection, fluctuations in groundwater level depth directly affect regional ecological patterns. Excessively low water levels may lead to wetland desiccation and biodiversity loss, while rapid rises can cause soil salinization and vegetation degradation (Singh et al., 2012). Relevant studies have also practically validated the significance of groundwater level prediction. For example, Liu et al. (2022) demonstrated in the lower Tarim River that machine learning–based groundwater level prediction models can quantitatively reveal current and future groundwater changes, clarifying the critical role of 'ecological water conveyance' in regional ecological restoration. Therefore, in-depth identification of the controlling mechanisms behind groundwater level depth variations and achieving high-precision spatiotemporal simulation are of great significance for promoting sustainable groundwater resource utilization and ecological environment protection."

Lines 62–67: The key disadvantage of physical models, compared to ML models, lies in their time-consuming setup, calibration, and validation processes. However, physical models have the advantage of offering more mechanistic insight into underlying hydrological processes, which ML models often lack.

Response: We thank the reviewer for highlighting the insufficient discussion on the comparison between physical models and machine learning models in the current manuscript. In response to your suggestion, we have revised and supplemented the relevant content accordingly. In the updated version, we have clearly stated the advantages of physical models in revealing the physical mechanisms of hydrological processes, while also acknowledging their limitations in regions with complex geological conditions due to high modeling complexity and substantial data requirements. The revised content is as follows:

"Current models used for simulating groundwater level dynamics can generally be categorized into two groups: physical models and machine learning models (Ao et al., 2021). Most physical models are based on hydrodynamic processes and water balance principles, and are capable of accurately representing the physical mechanisms of groundwater systems. Therefore, they possess irreplaceable advantages in characterizing groundwater flow and uncovering hydrological processes such as recharge, runoff, and discharge. However, in areas with complex geological structures or highly heterogeneous aquifer systems, the construction, parameter calibration, and validation of physical models typically require large amounts of high-resolution geological, hydrological, and hydraulic data. These requirements make physical modeling challenging to implement and time-consuming (Raghavendra N and Deka, 2014)."

Line 118: The model would benefit from incorporating a wider range of influencing factors, such as aquifer properties, topography, hydraulic conditions (e.g., lateral flow, vertical leakage, groundwater storage, surface water interactions), and anthropogenic variables like population density. Spatial heterogeneity in

evapotranspiration and precipitation should also be considered to improve model realism.

Response: We sincerely thank the reviewer for the professional and insightful comments. We fully agree that a variety of natural and anthropogenic factors—such as aquifer properties, topography, groundwater dynamics, population density, and evapotranspiration—can exert significant influence on regional groundwater level changes.

However, considering the design rationale and actual data availability in this study, we have carefully reflected on and responded to this point from the following two perspectives:

First, the core framework of our study is to independently construct an LSTM model for each monitoring well to simulate the temporal variation of groundwater level at that specific location. The model uses historical meteorological variables and anthropogenic dynamic factors (including air temperature, precipitation, snow depth, and groundwater extraction) as inputs, aiming to capture the nonlinear response relationship between these temporally dynamic factors and groundwater level changes. Under this modeling strategy, spatially static attributes such as aquifer properties and topography remain constant over short periods at a given site and thus cannot provide dynamic explanatory power for the temporal evolution of groundwater levels at that point. Additionally, the spatial heterogeneity of factors such as evapotranspiration and precipitation primarily influences regional-scale patterns or spatial distributions. Since our study focuses on site-specific time series modeling and identification of dominant influencing factors, it is relatively less dependent on spatially heterogeneous variables. We have clarified this limitation in Section 3.5 "Model Limitations" of the revised manuscript.

Second, regarding the absence of variables related to groundwater dynamics (e.g., lateral flow, vertical leakage, and surface–groundwater interactions), we fully acknowledge their critical roles in groundwater system evolution. Although in theory,

groundwater flow fields could be constructed through spatial interpolation of observed water levels, in our study the groundwater level is the target output variable of the model. Thus, prior to obtaining the model predictions, it cannot serve as an input driver. Moreover, in practice, there is a lack of independent observational data (such as hydraulic gradients or recharge–discharge rates) that directly reflect groundwater dynamics, making it currently unfeasible to incorporate these factors into the model. In future work, if data availability improves, we intend to include such variables as key supplementary inputs to enhance the model's physical interpretability.

Figure 2: Consider including a geological map that shows the distribution of geological formations or aquifer types. This would help contextualize the results spatially.

Response: We thank the reviewer for the suggestion. In response, we have added a new subfigure to Figure 2 showing the distribution of the aquifer system. The revised figure is as follows:

[Figure]

Spatial distribution of the ground surface elevation (a), topography (b) and aquifer system (c) in the Songnen Plain, China.

Figure 4: The observed and simulated groundwater levels do not align well; the simulated series appears overly variable. Please explain the possible causes of this discrepancy, such as overfitting, lack of key input variables, or limitations in the model's temporal resolution.

Response: We thank the reviewer for the valuable comments. Although the original manuscript included an explanation for the poor model performance at certain monitoring wells, the reasoning lacked clarity and failed to accurately convey the sources of model error. In response, we have revised and reorganized the relevant

paragraph to enhance its logical structure and coherence. The modified version is as follows:

"Only 18.11% of the monitoring wells in the study area had a Nash-Sutcliffe Efficiency (NSE) below 0.7 on the test dataset, and these wells were primarily located in the southern part of the western low plain (Figure 3a). In this region, the average absolute error between simulated and observed daily groundwater level depth ranged from 0.04 to 2.93 meters, although the maximum error reached as high as 11.56 meters (Figure 3c), indicating that the model exhibited certain instability in localized areas. Figure 4 compares the simulated and observed groundwater level depth series at several poorly performing wells in this region. As shown in the figure, significant discrepancies occurred during certain periods, and the fitting performance was unsatisfactory. The primary reason for this discrepancy is the large annual fluctuation in groundwater level depth at many wells in this region: 21.43% of the monitoring wells had a fluctuation range exceeding 10 meters. These extreme fluctuations posed challenges for the LSTM model's simulation accuracy. In the training data used for the LSTM model, samples with extreme values of groundwater level depth were relatively scarce, while samples with moderate values were more abundant. Consequently, the model tended to fit the data in the moderate range more accurately, resulting in limited predictive ability for the extreme ends of the groundwater level series. Despite the reduced accuracy at certain wells, the LSTM model is capable of accurately capturing the variation trend of groundwater levels, and no significant lag is observed between the simulated and observed values (Figure 4). The Pearson correlation coefficients at the four representative monitoring wells shown in the figure are 0.86, 0.81, 0.87, and 0.85, respectively. Moreover, the correlation coefficients reach their maximum values without applying any time lag, indicating that the simulated values can effectively and promptly reflect the actual variation trend of groundwater levels."

Lines 373–376 and 557–558: These sections are overly descriptive. Instead of

simply stating observations, clarify what the results reveal about the status or trends of water resources. Quantitative insights or implications for water management should be emphasized.

第 373–376 行与第 557–558 行：这两部分的描述偏于叙述性，应进一步挖掘这些结果对水资源状况或趋势的揭示意义。建议突出定量分析的结论或对水资源管理的启示。

Response: We sincerely thank the reviewer for the valuable comments. We have carefully revised the relevant sections of the manuscript.The main adjustments are summarized as follows:

1. Lines 373–376 (Analysis of frozen-thawed period dynamics):

While retaining the proportions and temporal characteristics of the three types of groundwater level dynamics during the frozen-thawed period, we further elaborated on the practical significance of their spatial differences for regional water resource management. We highlighted the indicative role of each dynamic type in reflecting groundwater recharge or depletion conditions and proposed a framework for differentiated zoning and adaptive regulation strategies based on these types to enhance the scientific basis and precision of groundwater management. The revised paragraph is as follows:

" Freeze-thaw processes intensify the transformation between soil water and groundwater (Daniel and Staricka, 2000; Miao et al., 2017; Lyu et al., 2022). As a typical seasonally frozen soil region, the Songnen Plain in China exhibits three main types of groundwater level fluctuations during the frozen-thawed period: 'decline during freezing and rise during thawing,' 'continuous decline,' and 'continuous rise' (Figure 6). The proportions of monitoring wells corresponding to these three types are 38.4% (V-shaped), 23.2% (continuously declining), and 38.4% (continuously rising), respectively. The distribution of these types reflects the diverse responses of regional groundwater systems to seasonal freeze-thaw processes, providing a foundation for refined water resource management. On one hand, this classification can help identify

potential recharge and depletion zones during spring, serving as a basis for groundwater storage adjustment, agricultural irrigation, and water resource allocation. On the other hand, it supports the development of dynamic management strategies tailored to freeze-thaw processes, enhancing responsiveness to groundwater level fluctuations in cold regions."

2. Lines 557–558 (Analysis of annual-scale dynamics):

We briefly summarized the differences in the controlling mechanisms of the three types of annual-scale groundwater dynamics. While precipitation is the dominant recharge source across the study area, the discharge pathways differ significantly. From the perspectives of both "naturally dominated" and "human activity dominated" processes, we proposed localized management strategies—such as enhancing ecological water use security and strengthening groundwater extraction control—emphasizing the value of classification results in improving the adaptability of groundwater resource regulation. The revised paragraph is as follows:

"The classification of annual-scale groundwater level dynamics identified three types: precipitation infiltration–evaporation type(29.0%), precipitation infiltration–runoff type(18.1%), and extraction type (52.9%). These results indicate that the regional groundwater system is generally controlled by precipitation recharge; however, the different types reflect distinct water level response mechanisms. The evaporation and runoff types are dominated by natural processes and exhibit groundwater fluctuations that are more sensitive to climatic conditions. In contrast, the extraction-driven type is associated with intensive groundwater use and is more responsive to changes in anthropogenic activities. These classification results provide a basis for tailored management. In areas dominated by natural processes, efforts should focus on securing ecological water demand and integrating rainwater resources to maintain groundwater system stability. In regions where extraction dominates, optimizing groundwater abstraction and improving water use structure are essential to mitigating continuous water level decline and enhancing resource sustainability."

---

## Author Comment (AC2)

Li et al. present an interesting study that employs deep learning models to predict groundwater levels during freezing and thawing periods, as well as to classify the underlying dynamic drivers. The paper is mostly well-written. Still, considerable issues require significant revision to make the paper clearer. The most important ones are related to the current structure; the results and discussion are in the same chapter, which is recommended for modification. The authors are encouraged to include a separate discussion section to discuss the main groundwater level types and the most significant implications from these different types. Second, some issues should be more precisely defined in the method. Finally, the authors should consider to present the conclusion in a more structured and clear way.

Response: We sincerely thank you for your recognition of our work and for your valuable comments. We have carefully studied your suggestions and made corresponding revisions to the manuscript. Regarding your comment about merging the "Results" and "Discussion" sections, we would like to clarify that in our study, the LSTM-based simulation results are closely integrated with the explanation provided by the Expected Gradients (EG) method. The two components are interdependent and difficult to present completely separately; thus, we initially adopted a combined section format to maintain content coherence and logical consistency. That said, we fully understand that a standalone "Discussion" section facilitates a deeper interpretation of the scientific significance. In response, we have added more discussion content in the relevant part of the manuscript, enhancing the analysis of the major groundwater level dynamics and their critical implications by incorporating insights from previous studies. These additions aim to improve the depth and academic value of the paper. Additionally, in response to your comment about the inaccuracy in the methodology section, we have thoroughly reviewed and revised the relevant descriptions to ensure greater precision and scientific rigor. For the conclusion section, we have also optimized its structure following your suggestion, making the summary clearer, more organized, and better aligned with the key findings and contributions of our study. Once again, we sincerely

thank you for your detailed comments and insightful suggestions, which have significantly contributed to improving the quality of this manuscript. We look forward to your further guidance.

General comments:

1. Abstract:

It would be worth rephrasing to make the message clear and better reflect the key findings and the value of this study.

Response: We sincerely thank the reviewer for the valuable comments on the abstract. We have revised the abstract accordingly. The revised version is as follows:

"Accurately characterizing groundwater level dynamics in seasonal frozen soil regions is of great significance for water resource management and ecosystem protection in cold areas. Taking the Songnen Plain in China as the study area, this paper constructs a Long Short-Term Memory (LSTM) model to simulate daily groundwater levels for 138 monitoring wells. The Expected Gradients (EG) method is introduced to interpret the model results, thereby identifying the dominant factors and underlying mechanisms of different groundwater level variation types. The results show that the LSTM model performs well on the test set, with the Nash-Sutcliffe Efficiency (NSE) exceeding 0.7 at 81.88% of the monitoring sites, effectively capturing the temporal dynamics of groundwater levels. At the annual scale, three typical groundwater level variation types are identified: precipitation infiltration–evaporation type (29.0%), precipitation infiltration–runoff type (18.1%), and extraction type (52.9%). The first two types are mainly controlled by natural processes, with water level variations depending on climatic conditions, while the extraction type is significantly influenced by human activities, characterized by frequent water level fluctuations. During the frozen-thaw period, groundwater level dynamics can be classified into three major types: "V"-shaped variation (decline during freezing, rise during thawing, accounting for 38.4%), continuous decline (23.2%), and continuous rise (38.4%). EG analysis indicates that the "V"-shaped dynamics are mainly governed by climatic factors such

as air temperature, precipitation, and snow thickness, clearly reflecting the dominant role of the frozen-thaw process. Further analysis reveals that when the initial groundwater level depth at the start of the freezing period is shallower than the sum of the "frozen-thaw influence depth plus capillary rise height," a hydraulic connection is established between the frozen soil layer and the aquifer, enabling frequent conversion between soil water and groundwater and resulting in the characteristic "V"-shaped fluctuation. Conversely, when the groundwater level depth exceeds this critical threshold, the frozen-thaw process has limited influence on the aquifer. Groundwater level variations are then mainly driven by groundwater extraction or the recovery process following prior recharge from precipitation, exhibiting continuous decline or continuous rise, respectively. This study establishes an integrated framework of "simulation–classification–interpretation," which not only improves the accuracy of groundwater level dynamic simulation and prediction but also provides new methods and perspectives for revealing the underlying mechanisms. The findings offer theoretical support and technical basis for regional groundwater resource management, regulation strategy optimization, and climate change response assessment in cold regions."

2. Method:

a) Figure 2 What do the solid circles in Fig. 2(a) represent? Additional description on these labels should be added to the figure caption. Since some similar information is presented in panels (a), (b), and (c), consider merging some of them.

Response: We thank the reviewer for the comment. Regarding the solid circles in Figure 2(a), we have confirmed that this marker was mistakenly added during the typesetting process and has been removed in the revised version. In addition, in response to the suggestion concerning the redundancy of certain information across the subplots in Figure 2(a), (b), and (c), we have merged and adjusted the figure contents accordingly. The revised figure is shown below:

[Figure]

Spatial distribution of the ground surface elevation (a), topography (b) and aquifer system (c) in the Songnen Plain, China.

b) Lines 147-149 How do you determine the exact timing of the beginning and end of the freezing period for each well? A precise definition of the freezing period should be provided, similar to the one you gave for the 'Beginning of winter' in Lines 194-198.

Response: Thank you very much for your comments. Based on meteorological data from 2018 to 2021 and relevant studies (Lyu et al., 2023), we found that after the "Beginning of Winter" solar term (around November 7–8), air temperatures steadily declined and a thin ice layer began to form on the ground surface. Following the "Rain

Water" solar term (around February 18–20), temperatures began to rise, and the frozen soil started to gradually thaw. By the "Grain Rain" solar term (around April 19–21), most areas in the study region had experienced complete thawing of the frozen soil. Accordingly, we defined a uniform freezing and thawing period for all monitoring wells in the study area. Specifically, the freezing period is defined as the time span from "Beginning of Winter" to "Rain Water", and the thawing period is from "Rain Water" to "Grain Rain" each year. We will provide additional clarification in the manuscript regarding the start and end times of the freezing and thawing periods.

c) Lines 167-169 Please detail the method to estimate the groundwater extraction volume. Given that the groundwater extraction volume is a key component of the proposed mechanism, its estimation accuracy may have an impact on the results. Also, the well depth and screened interval of the monitoring wells might also influence the response rate of the observed groundwater levels, but this aspect does not appear to be addressed in the paper.

Response: Thank you very much for your comments. In the Songnen Plain, approximately 70% of groundwater extraction is used for agricultural irrigation; therefore, in this study, groundwater extraction was approximated based on crop water deficits. Using spatial distribution data of the region's major crops, ten-day period crop water requirements, and precipitation data, we estimated groundwater extraction at a fine resolution, ultimately generating ten-day period groundwater extraction data with a spatial resolution of 25 km × 25 km. Specifically, based on the water requirements of the main crops (rice, soybean, and maize), we calculated the total crop water demand for each ten-day period within each grid cell. These values were then weighted according to the crop planting area to obtain the total water demand per grid. By comparing precipitation with crop water demand, we determined whether precipitation could meet the crop water needs. When precipitation was sufficient, crops relied entirely on natural rainfall, and the effective precipitation equaled the water demand. When precipitation was insufficient, effective precipitation was limited by actual

rainfall, and the remaining crop water deficit was assumed to be supplemented by other water sources. Finally, the difference between crop water demand and effective precipitation was calculated as the crop water deficit, which was assumed to be primarily supplied by groundwater. This allowed us to approximate ten-day period groundwater extraction. To ensure consistency with the temporal resolution of other variables used for model training, the ten-day period data were converted to daily averages by dividing by the number of days in each period. We will provide additional clarification in the manuscript regarding the method used to estimate groundwater extraction.

We fully acknowledge that the depth of monitoring wells and the distribution of screened intervals may affect the groundwater level response. However, due to limited data availability, we were unable to obtain relevant structural parameters for all wells and thus could not conduct a detailed analysis in this study. In future work, we will further explore the impact of these factors when sufficient data become available.

3. Result and discussion:

a) Lines 313-314 This statement is unclear or lacks significance. Could you provide a quantitative indicator to support it?

Response: We thank the reviewer for the comment. As suggested, we have revised the sentence accordingly. The updated version is as follows:

"The LSTM model is capable of accurately capturing the variation trend of groundwater levels, and no significant lag is observed between the simulated and observed values (Figure 4). The Pearson correlation coefficients at the four representative monitoring wells shown in the figure are 0.86, 0.81, 0.87, and 0.85, respectively. Moreover, the correlation coefficients reach their maximum values without applying any time lag, indicating that the simulated values can effectively and promptly reflect the actual variation trend of groundwater levels."

b) Lines 359 The authors are strongly recommended to label the three monitoring wells representing the three types of groundwater level dynamics (panels b, c, and d) in

Figure 5a. The well numbers mentioned here are not very informative since the locations of the wells are not indicated. There are similar cases later on as well.

Response: We thank the reviewer for the suggestion. We have marked the locations of the three representative monitoring wells in Figure 5a. In addition, the locations of the representative wells have also been added to Figure 6. The revised figures are as follows:

[Figure]

(a) Spatial distribution of different annual groundwater level dynamic types in the Songnen Plain, China; (b–d) Dynamic curves of different annual groundwater types and their corresponding precipitation variations. (b) The first annual dynamic type is represented by an unconfined aquifer monitoring well, numbered 230204210070, located in the western low plain; (c) The second annual dynamic type is represented by an unconfined aquifer monitoring well, numbered 220182210411, located in the Lasong Block between rivers; (d) The third annual dynamic type is represented by an unconfined aquifer monitoring well, numbered 220802210145, located in the western piedmont sloping plain.

[Figure]

(a–c) Dynamic curves of different groundwater types during the freeze–thaw period and corresponding changes in air temperature; (d) Spatial distribution of different groundwater level dynamic types during the freeze–thaw period in the Songnen Plain, China. The dynamic curves of the groundwater level exhibiting patterns of (a) V-shaped, (b) continuous decline, and (c) continuous rise correspond to the unconfined aquifer monitoring wells numbered 230204210070, 220182210411, and 220802210145, respectively.

c) Lines 388-395 I am not sure I fully understand the authors' meaning here. They state that continuous groundwater level decline mostly occurs in areas with deep groundwater level, but actually, the groundwater depth is greater in areas where the groundwater level shows a continuous rise. Moreover, I think some of the mechanism for the "continuous rising" type should be discussed further, that could enhance the implication of this study.

Response: We sincerely thank the reviewer for the constructive suggestion. The original statement that "sustained declines in groundwater levels mostly occur in areas with greater groundwater depths" could indeed be misleading, as some areas with sustained rising trends actually have even deeper groundwater levels. We have revised and rephrased this section The revised content is as follows:

"Monitoring points with the continuous decline in the groundwater level were mainly distributed in areas, such as the eastern high plain and the Lasong Block between rivers, where the groundwater level depth ranged from 4.52 to 11.51 m at the start of the freezing period (Fig. 6d)."

In addition, we have further discussed the mechanism underlying the formation of the "sustained rising" groundwater level trend. The revised content is as follows:

"Groundwater monitoring points exhibiting the precipitation infiltration-runoff type were mainly distributed in the eastern high plain and the Lasong Block between rivers. In these areas, the groundwater level is deeper, typically ranging from 5 to 12 m (Fig. 11b), and runoff is the primary mode of groundwater discharge. The deeper groundwater level prolongs the infiltration time of precipitation, resulting in a delayed response of the groundwater level dynamics to precipitation recharge. Groundwater level peaks typically occur between August and October (Fig. 11d), lagging behind the precipitation peak by approximately one month (Fig. 11f). Due to the low recharge rate, groundwater level fluctuations are relatively moderate, with annual variations generally within 4 m (Fig. 11c). During the freeze–thaw period, groundwater monitoring points with continuously declining trends have greater initial groundwater level depths, ranging from 4.52 to 11.51 m at the beginning of the freezing period (Fig. 12d). This pattern is primarily attributed to the sharp reduction in groundwater extraction following the end of the irrigation season. As agricultural activities cease, the regional groundwater system gradually enters a recovery phase, during which the groundwater depression cones formed by intensive earlier pumping begin to be replenished, leading to a gradual rise in groundwater levels. Due to the previously high pumping intensity and the relatively deep groundwater table, the recovery process does not occur instantaneously; instead, it is jointly constrained by the delayed response of the groundwater system and the regional recharge conditions. As a result, the groundwater level exhibits a steady and sustained upward trend. In addition, the soil freezing depth in this dynamic type was shallower (between 1.6 and 1.8 m), and the soil was still

primarily silty clay (Fig. 12b and c). The greater groundwater level depth and shallower soil freezing depth prevented a complete hydraulic connection between the frozen soil and groundwater (Fig. 12a), resulting in the groundwater level being unaffected by the soil freeze–thaw process. Therefore, under conditions where no groundwater extraction occurs during the freeze–thaw period and the groundwater level is not influenced by freeze–thaw processes, the groundwater system continues the post-irrigation recovery process, presenting a "sustained rising" groundwater level pattern."

d) Line 427 It is confusing to see the sentence "Precipitation directly recharged the groundwater" here.

Response: We thank the reviewer for pointing out the issue with the statement "Precipitation directly recharged the groundwater." We acknowledge that this expression was logically ambiguous and lacked terminological rigor. We have revised the sentence by linking it more clearly to the preceding one. The revised version is as follows:

"When a pronounced precipitation peak occurred (Figure 9b), the EG score increased significantly (exceeding 0.15), corresponding to a rise in groundwater level (Figure 9e), indicating that precipitation infiltration made a substantial contribution to the groundwater level increase."

e) Some subheadings are a bit too long and very similar, e.g., Sections 3.2, 3.2.1, and 3.2.2, as well as 3.3, 3.3.1, and 3.3.2, I suggest the authors refine them.

Response: We thank the reviewer for the suggestion. We have simplified and refined the subheadings of Sections 3.2 and 3.3 by removing redundant words and highlighting the core content. The revised subheadings are as follows:

3.2. Dynamic Characteristics of Regional Groundwater Level and their Distribution Laws

3.2.1. Annual Dynamics Variations and Spatial Distribution

3.2.2. Freeze–Thaw Period Dynamics Variations and Spatial Distribution

3.3. Main Controlling Factors and Identification of Causes for Various

Groundwater Level Dynamic Types

3.3.1. Annual Dynamics: Influencing Factors and Dynamics Mechanisms

3.3.2. Freeze–Thaw Dynamics: Influencing Factors and Dynamics Mechanisms

f) The authors are encouraged to strengthen the discussion by connecting this research to relevant studies and highlighting its potential implications.

Response: In response to the reviewer's suggestion, we have strengthened the discussion by incorporating relevant existing studies to better support our conclusions and highlight the practical value of this research. The specific revisions include:

In the analysis of the first intra-annual groundwater dynamics type, we have added a citation to the findings of Xu et al. (2024) in the Songnen Plain, which demonstrated that precipitation is the primary source of shallow groundwater recharge. This indirectly supports our proposed "precipitation infiltration–evaporation" mechanism.

In the analysis of the third intra-annual dynamics type, we have included a reference to the study by Wu et al. (2025) on groundwater level variations in the Songnen Plain, which pointed out that significant groundwater declines are mainly related to excessive agricultural extraction—particularly in large-scale rice cultivation areas in Jilin Province. This finding is highly consistent with our identified "extraction-driven" mechanism.

4. Conclusion:

The conclusion section is considerably longer than necessary and could be more concise.

Response: We sincerely thank the reviewer for the valuable suggestions regarding the conclusion. In response, we have revised the conclusion as follows:

"This study applies an interpretable deep learning approach to reveal the driving mechanisms behind groundwater level dynamics in seasonally frozen soil regions. High-precision simulations were conducted at 138 monitoring wells using an LSTM model. The main controlling factors and underlying mechanisms of different

groundwater level variation types were identified using the EG (Expected Gradients) method. The main findings are as follows:

First, the LSTM model demonstrated high accuracy in simulating groundwater level variations in seasonally frozen areas, with NSE values on the test set ranging from 0.53 to 0.96, indicating its effectiveness in capturing complex groundwater dynamics.

Second, by applying the EG method, three dominant intra-annual groundwater dynamic types in the Songnen Plain of China were identified: precipitation infiltration–evaporation type (29.0%), precipitation infiltration–runoff type (18.1%), and extraction type (52.9%). Correspondingly, during the freeze–thaw period, these types are reflected as V-shaped, continuously declining, and continuously rising patterns, accounting for 38.4%, 23.2%, and 38.4% of the monitoring wells, respectively.

Third, while all three intra-annual types are primarily recharged by precipitation infiltration, their discharge pathways differ: evaporation, surface runoff, and groundwater extraction, respectively. During the freeze–thaw period, changes in the soil water potential gradient due to freezing and thawing lead to interactions between soil water and groundwater, resulting in the V-shaped variation. In contrast, the continuously declining and rising types reflect gradual water level changes primarily driven by groundwater extraction and precipitation recharge, without strong influence from freeze–thaw processes. These dynamic types represent groundwater fluctuations jointly driven by multiple factors across different temporal scales.

Groundwater dynamics in seasonally frozen regions are complex, influenced by both climate variability and human activities. Deep learning models require more sophisticated architectures and broader input variables to improve simulation accuracy, but this increases the difficulty of interpreting their internal mechanisms. Therefore, this study introduces the EG method to identify the key drivers and formative mechanisms of groundwater level dynamics. The results demonstrate the great potential of the EG method to bridge model accuracy and interpretability, offering a new perspective for analyzing complex hydrological processes. Future research may

incorporate more advanced interpretability techniques to further enhance understanding of deep learning models. The significance of deep learning lies not only in high-accuracy simulations, but also in advancing the discovery of hydrological mechanisms. This study provides new methodological support and theoretical insights for groundwater resource management and ecological protection in seasonally frozen soil regions."

Minor comments:

Line 49 There are formatting issues with some references, which also appear throughout the rest of the paper.

Response: We sincerely thank the reviewer for the review. We have conducted a comprehensive check of all references cited in the manuscript and have standardized their formatting in accordance with the journal's guidelines to ensure accuracy and consistency.

Line 137 delete "topography of the"

Response: Thank you for the suggestion. We have removed "topography of the" in Line 137 to make the expression more accurate and concise.

I'm not sure if it's due to image resolution, but some of the colors in the figures are difficult to distinguish. For example, in Fig. 2a, the colors of the solid circles are too similar to those used in the base map.

Response: We thank the reviewer for the comment. Regarding the solid circles in Figure 2(a), we have confirmed that this marker was mistakenly added during the typesetting process and has been removed in the revised version. In addition to Fig. 2(a), we also noticed a similar issue with insufficient color contrast in Fig. 11(a) of the original manuscript. To improve the readability and visual clarity of the figure, we have adjusted the color of the solid circles in Fig. 11(a) to enhance their contrast against the background map and minimize potential misinterpretation.

**References:**

Xu, L., Cui, X., Bian, J., et al.: Dynamic change and driving response of shallow groundwater level based on random forest in southwest Songnen Plain, J. Hydrol. Reg. Stud., 53, 101800, https://doi.org/10.1016/j.ejrh.2024.101800, 2024.

Wu, H., Ye, X., Du, X., et al.: Assessing groundwater level variability in response to climate change: A case study of large plain areas, J. Hydrol. Reg. Stud., 57, 102180, https://doi.org/10.1016/j.ejrh.2025.102180, 2025.

---

## Author Comment (AC3)

The groundwater level changes in the seasonal frozen soil region are simulated using interpretable deep learning, while the underlying mechanisms of groundwater level dynamics during the freezing and thawing periods as well as non-freezing and thawing periods are revealed. The topic is interesting and the research results can provide a reference for the assessment of groundwater resources in seasonal frozen soil regions. However, considering that the Hydrology and Earth System Sciences is the world's premier journal publishing research of the highest quality in hydrology, it could not be accepted before a minor revision.

Response: Thank you very much for your appraisal of our work and encouragement. We have studied comments carefully and have made correction which we hope meet with approval.

1)During the freeze–thaw process, the groundwater level exhibits a noticeable lag during the recovery phase. Has the author considered the physical mechanisms behind this lag, such as delayed soil thawing or the blockage effect of frozen layers?

Response: We sincerely thank the reviewer for the insightful and professional comments. As noted, we indeed observed a significant lag between the rise in air temperature and the corresponding rise in groundwater levels during the freeze–thaw period. Our analysis suggests that this phenomenon is closely related to the staged thawing process of the frozen soil in the study area and its hindering effect on vertical water movement.

Specifically, based on meteorological and soil temperature monitoring data from 2018 to 2021 in the study area, the thawing period can be roughly divided into three stages. In the first stage (around late February), air temperature first rises above 0 °C, initiating snowmelt; however, due to diurnal freeze–thaw cycles, the frozen soil has not yet completely thawed, and infiltration of liquid water is impeded. In the second stage (around mid to late March), air temperature remains steadily above 0 °C, the frozen soil gradually thaws through, and water begins to infiltrate into the aquifer, leading to a more rapid rise in groundwater levels. In the third stage (around mid-April), the frozen

soil is completely thawed, and vertical water pathways become fully open. Due to the staged nature of frozen soil thawing, during the early phase of air temperature increase—i.e., between the first and second stages—although snowmelt water is present, residual frozen layers within the soil profile form a distinct "water-blocking layer" that inhibits downward infiltration and aquifer recharge. Even if liquid water appears in the shallow soil, it cannot directly recharge the groundwater. Only after the complete thawing of the frozen soil can continuous vertical flow pathways be established, allowing for a noticeable rise in groundwater levels.

Based on existing data, this study provides a preliminary explanation of the observed lag phenomenon. In future work, we plan to incorporate parameters such as frozen soil thickness, hydraulic properties, and soil profile structure to conduct quantitative analysis and further elucidate the response mechanisms of groundwater dynamics to freeze–thaw processes.

2)Line 314 mentions that "there is no significant lag between the simulated and observed values." Has any correlation or lag correlation analysis been conducted to support this statement?

Response: We thank the reviewer for the valuable comments. In response to the statement regarding the absence of an obvious lag between the simulated and observed values, we have added Pearson correlation analysis for the four representative monitoring wells shown in Figure 4. The revised content is as follows:

"The LSTM model is capable of accurately capturing the variation trend of groundwater levels, and no significant lag is observed between the simulated and observed values (Figure 4). The Pearson correlation coefficients at the four representative monitoring wells shown in the figure are 0.86, 0.81, 0.87, and 0.85, respectively. Moreover, the correlation coefficients reach their maximum values without applying any time lag, indicating that the simulated values can effectively and promptly reflect the actual variation trend of groundwater levels."

3)Line 211 states that 150 days of meteorological variables were used as model

input. What is the basis for selecting this window length? Has other time lengths been tested for their effect on model performance?

Response: We thank the reviewer for raising this professional and important question. The choice of a 150-day input window for meteorological variables was based on the following considerations:

First, the effects of meteorological changes on groundwater dynamics often require a relatively long period of transmission and accumulation, which short windows may fail to capture adequately. Based on a time-series analysis of long-term groundwater and meteorological data in the Songnen Plain, we found that a span of approximately five months (150 days) can effectively reflect the combined influence of temperature, precipitation, and other variables on groundwater level fluctuations.

Second, we conducted comparative experiments using different window lengths of 90, 120, 150, and 180 days for model training and testing. The results showed that the 150-day window outperformed the others in terms of root mean square error (RMSE), coefficient of determination ($R^2$), and Nash–Sutcliffe efficiency (NSE). It also avoided the computational inefficiency and potential overfitting issues associated with excessively long windows.

Based on the above analysis, we selected a 150-day input window to strike a balance between physical interpretability and model performance, thereby ensuring the scientific validity and stability of the prediction results.

4)How is the early stopping strategy for the LSTM model set?

Response: To further improve training efficiency and effectively prevent overfitting, two strategies were adopted during the training of the LSTM model: early stopping and adaptive learning rate adjustment. The coefficient of determination ($R^2$) on the validation set was uniformly used as the performance evaluation metric.

For the early stopping strategy, training was automatically terminated when the improvement in $R^2$ on the validation set remained below a predefined threshold (0.01) for 30 consecutive training epochs, indicating that the model had converged. This

approach helps conserve computational resources and avoid overfitting. After termination, the model parameters were restored to the state corresponding to the epoch with the highest $R^2$ on the validation set, ensuring optimal generalization performance.

In addition, to enhance precision optimization in the later stages of convergence, an adaptive learning rate decay mechanism was introduced. Specifically, when the validation $R^2$ did not improve by more than 0.01 for 15 consecutive epochs, the current learning rate was automatically reduced to half of its previous value, slowing down parameter updates and improving the model's ability to search near the optimum. To prevent the learning rate from becoming too small and stalling the training process, a minimum learning rate limit was set at 1% of the initial value.

5)It is recommended to include comparison plots for typical sites with NSE > 0.7 in the test set, to contrast with the low-performance sites in Figure 4 (NSE < 0.7), and to further validate the model's applicability and stability across different locations.

Response: We thank the reviewer for the suggestion. As recommended, we have added comparison plots for typical sites with NSE > 0.7 in the test set, to contrast with the low-performance sites in Figure 4. The supplementary figure is as follows:

[Figure]

Comparison between simulated and observed groundwater table depths at typical sites in the study area with test set NSE > 0.7.

6)When using the EG method to calculate the importance of influencing factors, have you considered converting the EG scores into percentages to more clearly display the dominant factors and their relative contributions at different periods for the same groundwater level dynamic type?

Response: We thank the reviewer for the valuable suggestion. During our study, we did consider converting the EG scores into percentage form to provide a more intuitive representation of the relative importance of different variables. However, the EG method outputs a time series that reflects the influence of each input variable on the model's prediction at each time step. Even after converting the EG scores into percentages, they remain a time-dependent sequence rather than a single static value, making it difficult to comprehensively assess the overall importance of each variable

throughout the entire prediction period. Moreover, the importance of different variables varies significantly across different time periods. Simple normalization or averaging may obscure the contribution of certain variables during critical periods. Therefore, we chose to retain the original time series of EG scores and performed qualitative analyses based on representative time intervals, in order to better reflect the stage-specific dominant mechanisms affecting groundwater level dynamics.

7)The manuscript refers to the "initial groundwater level depth at the start of the freezing period." How is the time point of this variable consistently defined? Is it synchronized with the time when the maximum freezing depth occurs?

Response: The term "initial groundwater level depth at the beginning of the freezing period" in the manuscript refers to the groundwater level depth corresponding to the early stage of the freezing period each year—specifically, when air temperature consistently drops below 0 °C and surface freezing first occurs. To standardize the time point across different years and monitoring sites, we adopted the solar term "Lidong" (approximately November 7–8) as a unified indicator for the onset of the freezing period. The groundwater level depth on the day of Lidong was extracted and used as the initial groundwater level depth. This time point was determined based on the regional climatic patterns of the study area and does not coincide with the occurrence of the maximum freezing depth. According to previous studies and our local observations, the maximum freezing depth typically occurs in the later part of winter (e.g., from late January to mid-February), lagging behind the beginning of the freezing period.

8)In line 691, the conclusion states that a "V-shaped" groundwater level trend indicated a significant influence of the soil freeze–thaw process on the groundwater level. However, the specific causes of the V-shaped dynamics are not clearly explained.

Response: We sincerely thank the reviewer for the valuable suggestions. In response, we have revised the conclusion section as follows:

"During the freeze–thaw period, changes in the soil water potential gradient due

to freezing and thawing lead to interactions between soil water and groundwater, resulting in the V-shaped variation. In contrast, the continuously declining and rising types reflect gradual water level changes primarily driven by groundwater extraction and precipitation recharge, without strong influence from freeze–thaw processes."

9)In line 222, the formula should be revised to:$f_t = \sigma(W_{xf}x_t + W_{hf}h_{t-1} + b_f)$

Response: Thank you for pointing out the error in the formula. As suggested, we have corrected the formula in line 222 accordingly.

10)It is recommended to display the specific NSE value of the representative site in the western low plain region within the test set in Figure 4.

Response: We sincerely thank the reviewer for the valuable suggestion. We have added the specific NSE values for representative sites in the low western plain area within the corresponding section of Figure 4. The revised figure is as follows:

[Figure]

Comparison of the simulated and observed groundwater level depths at typical points

in the western low plain (NSE values on the test set < 0.7).

11)It is suggested to delete Figure 2d and merge Figure 2b with Figures 2a and 2c.

Response: We sincerely thank the reviewer for the valuable suggestion. In response, we have revised Figure 2 as suggested. The updated figure is as follows:

[Figure]

Spatial distribution of the ground surface elevation (a), topography (b) and aquifer system (c) in the Songnen Plain, China.

---

## Author Response (AR1)

Li et al. present an interesting study that employs deep learning models to predict groundwater levels during freezing and thawing periods, as well as to classify the underlying dynamic drivers. The paper is mostly well-written. Still, considerable issues require significant revision to make the paper clearer. The most important ones are related to the current structure; the results and discussion are in the same chapter, which is recommended for modification. The authors are encouraged to include a separate discussion section to discuss the main groundwater level types and the most significant implications from these different types. Second, some issues should be more precisely defined in the method. Finally, the authors should consider to present the conclusion in a more structured and clear way.

Response: We sincerely appreciate your recognition of our research work and your valuable suggestions. In response to the major concerns you raised, we have made corresponding revisions and improvements. Regarding the issue of merging the "Results" and "Discussion" sections, we have retained the overall logical coherence of the chapters while supplementing the discussion content. We have also clearly distinguished the discussion topics in the form of subsections to enhance the structure and readability of the manuscript. Specifically, we added "4.1 Implications of Groundwater Level Dynamics Classification for Water Resources Management" and "4.2 A New Perspective on Identifying Groundwater Level Dynamic Mechanisms" in the discussion section. For the inaccuracies in the description of the methodology, we have reorganized and precisely restated the relevant content. Finally, we have revised the "Conclusions" section to make it more concise and well-structured.

General comments:

1. Abstract:

It would be worth rephrasing to make the message clear and better reflect the key findings and the value of this study.

Response: We sincerely thank the reviewer for the valuable comments on the

abstract. We have revised the abstract accordingly. The revised content is located from line 14 to line 42.

2. Method:

a) Figure 2 What do the solid circles in Fig. 2(a) represent? Additional description on these labels should be added to the figure caption. Since some similar information is presented in panels (a), (b), and (c), consider merging some of them.

Response: We thank the reviewer for the comment. Regarding the solid circles in Figure 2(a), we have confirmed that this marker was mistakenly added during the drawing process and has been removed in the revised manuscript. In addition, in response to the suggestion concerning the redundancy of certain information across the subplots in Figure 2(a), (b), and (c), we have merged and adjusted the figure contents accordingly. The revised figure is shown below:

[Figure]

Spatial distribution of the ground surface elevation (a), topography (b) and aquifer system (c) in the Songnen Plain, China.

b) Lines 147-149 How do you determine the exact timing of the beginning and end of the freezing period for each well? A precise definition of the freezing period should be provided, similar to the one you gave for the 'Beginning of winter' in Lines 194-198.

Response: Thank you very much for your comments. In the original manuscript, lines 147–149 only provided an overview of the general conditions of soil freezing and thawing in the study area, without giving a precise definition of the start and end times of the freezing period for each monitoring well. According to your suggestion, we have added the start and end times of the freezing period for each monitoring well in the revised manuscript, lines 238–241.

c) Lines 167-169 Please detail the method to estimate the groundwater extraction volume. Given that the groundwater extraction volume is a key component of the proposed mechanism, its estimation accuracy may have an impact on the results. Also, the well depth and screened interval of the monitoring wells might also influence the response rate of the observed groundwater levels, but this aspect does not appear to be addressed in the paper.

Response: Thank you very much for your comments. We have provided additional explanations of the method for estimating groundwater extraction in the revised manuscript, lines 196–214. In this study, groundwater extraction was approximated using the crop water deficit method. This approach is based on crop planting area, crop water requirements, and precipitation data, which allows quantification of groundwater demand during the irrigation period and is therefore theoretically justified. To assess the reliability of the method, we compared the annual total crop water deficit with the annual groundwater supply in local areas. The results show a strong correlation between the two, and the interannual variation of total crop water deficit is consistent with that of groundwater supply, supporting the effectiveness of this method in reflecting groundwater extraction.

We fully acknowledge that the well depth and screened interval of the monitoring

wells may affect the groundwater level response. However, due to limited data availability, we were unable to obtain relevant structural parameters for all wells and thus could not conduct a detailed analysis in this study. In future work, we will further explore the impact of these factors when sufficient data become available.

3. Result and discussion:

a) Lines 313-314 This statement is unclear or lacks significance. Could you provide a quantitative indicator to support it?

Response: We thank the reviewer for the comment. To address the lack of clarity in lines 313–314 of the original manuscript, we computed the Pearson correlation coefficients at lags of 0–7 days between the simulated and observed water levels for the four representative monitoring wells shown in Figure 4, and added the corresponding quantitative indicators in the revised manuscript, lines 358–363.

b) Lines 359 The authors are strongly recommended to label the three monitoring wells representing the three types of groundwater level dynamics (panels b, c, and d) in Figure 5a. The well numbers mentioned here are not very informative since the locations of the wells are not indicated. There are similar cases later on as well.

Response: We thank the reviewer for the suggestion. We have marked the locations of the three representative monitoring wells in Figure 5a. In addition, the locations of the representative wells have also been added to Figure 6. The revised figures are as follows:

[Figure]

(a) Spatial distribution of different annual groundwater level dynamic types in the Songnen Plain, China; (b–d) Dynamic curves of different annual groundwater types and their corresponding precipitation variations. (b) The first annual dynamic type is represented by an unconfined aquifer monitoring well, numbered 230204210070, located in the western low plain; (c) The second annual dynamic type is represented by an unconfined aquifer monitoring well, numbered 220182210411, located in the Lasong Block between rivers; (d) The third annual dynamic type is represented by an unconfined aquifer monitoring well, numbered 220802210145, located in the western piedmont sloping plain.

[Figure]

(a–c) Dynamic curves of different groundwater types during the freeze–thaw period and corresponding changes in air temperature; (d) Spatial distribution of different groundwater level dynamic types during the freeze–thaw period in the Songnen Plain, China. The dynamic curves of the groundwater level exhibiting patterns of (a) V-shaped, (b) continuous decline, and (c) continuous rise correspond to the unconfined aquifer monitoring wells numbered 230204210070, 220182210411, and 220802210145, respectively.

c) Lines 388-395 I am not sure I fully understand the authors' meaning here. They state that continuous groundwater level decline mostly occurs in areas with deep groundwater level, but actually, the groundwater depth is greater in areas where the groundwater level shows a continuous rise. Moreover, I think some of the mechanism for the "continuous rising" type should be discussed further, that could enhance the implication of this study.

Response: We sincerely thank the reviewer for the constructive suggestion. The original statement that "sustained declines in groundwater levels mostly occur in areas with greater groundwater depths" could indeed be misleading, as some areas with sustained rising trends actually have even deeper groundwater levels. We have revised and rephrased this section The revised content is as follows, located on lines 435-438

of the revised manuscript:

"Monitoring points with the continuous decline in the groundwater level were mainly distributed in areas, such as the eastern high plain and the Lasong Block between rivers, where the groundwater level depth ranged from 4.52 to 11.51 m at the start of the freezing period (Fig. 6d)."

In Section 3.3.2, based on the results of the EG method, we explicitly pointed out that the "continuously rising" groundwater level type is not affected by the freeze–thaw process, and its cause is mainly attributed to the intra-annual groundwater level recovery process. However, its corresponding intra-annual groundwater level dynamic type had not yet been identified. In Section 3.3, we clarified the correspondence between intra-annual and freeze–thaw period groundwater level dynamic types, indicating that the "continuously rising" type corresponds to the extraction type in intra-annual dynamics. Therefore, in the revised manuscript (lines 645-653), we have provided a more detailed explanation of the formation mechanism of the "continuously rising" groundwater level type.

d) Line 427 It is confusing to see the sentence "Precipitation directly recharged the groundwater" here.

Response: We thank the reviewer for pointing out the issue with the statement "Precipitation directly recharged the groundwater." We acknowledge that this expression was logically ambiguous and lacked terminological rigor. We have revised the sentence by linking it more clearly to the preceding one. The revised content is as follows, located on lines 471-474 of the revised manuscript:

"When a pronounced precipitation peak occurred (Figure 9b), the EG score increased significantly (exceeding 0.15), corresponding to a rise in groundwater level (Figure 9e), indicating that precipitation infiltration made a substantial contribution to the groundwater level increase."

e) Some subheadings are a bit too long and very similar, e.g., Sections 3.2, 3.2.1, and 3.2.2, as well as 3.3, 3.3.1, and 3.3.2, I suggest the authors refine them.

Response: We thank the reviewer for the suggestion. We have simplified and refined the subheadings of Sections 3.2 and 3.3 by removing redundant words and highlighting the core content. The revised subheadings are as follows:

3.2. Dynamic Characteristics of Regional Groundwater Level and their Distribution Laws

3.2.1. Annual Dynamics Variations and Spatial Distribution

3.2.2. Freeze–Thaw Period Dynamics Variations and Spatial Distribution

3.3. Main Controlling Factors and Identification of Causes for Various Groundwater Level Dynamic Types

3.3.1. Annual Dynamics: Influencing Factors and Dynamics Mechanisms

3.3.2. Freeze–Thaw Dynamics: Influencing Factors and Dynamics Mechanisms

f) The authors are encouraged to strengthen the discussion by connecting this research to relevant studies and highlighting its potential implications.

Response: Thank you for the valuable comments from the reviewers. Based on the suggestions, we have added a new "Discussion" section in the revised manuscript, focusing on Section 4.1 "Implications of Groundwater Level Dynamics Classification for Water Resources Management" and Section 4.2 "A New Perspective on Identifying Groundwater Level Dynamic Mechanisms." Section 4.1 further validates the accuracy of our study results by citing and comparing relevant existing research and discusses the application value of the study in detail. Section 4.2 mainly introduces the Expected Gradient (EG) method used in this study, explaining its differences and advantages compared to traditional research on groundwater level dynamic mechanisms. The revised content is located from lines 697 to 755.

4. Conclusion:

The conclusion section is considerably longer than necessary and could be more concise.

Response: We sincerely thank the reviewer for the valuable suggestions regarding

the conclusion. In response, we have revised the conclusion accordingly. The revised content is located from line 788 to line 820.

Minor comments:

Line 49 There are formatting issues with some references, which also appear throughout the rest of the paper.

Response: We sincerely thank the reviewer for the review. We have conducted a comprehensive check of all references cited in the manuscript and have standardized their formatting in accordance with the journal's guidelines to ensure accuracy and consistency.

Line 137 delete "topography of the"

Response: Thank you for the suggestion. We have removed "topography of the".

I'm not sure if it's due to image resolution, but some of the colors in the figures are difficult to distinguish. For example, in Fig. 2a, the colors of the solid circles are too similar to those used in the base map.

Response: We thank the reviewer for the comment. Regarding the solid circles in Figure 2(a), we have confirmed that this marker was mistakenly added during the drawing process and has been removed in the revised manuscript . In addition to Fig. 2(a), we also noticed a similar issue with insufficient color contrast in Fig. 11(a) of the original manuscript. To improve the readability and visual clarity of the figure, we have adjusted the color of the solid circles in Fig. 11(a) to enhance their contrast against the background map and minimize potential misinterpretation.

**Anonymous referee #2**

This manuscript applies a machine learning (ML) approach to predict time-varying groundwater levels in seasonally freezing regions of China. The topic is timely and of high importance for groundwater resource management and environmental protection. However, the study overlooks several critical factors that could significantly influence the results and interpretations. By incorporating additional hydrogeological and environmental variables, the model's accuracy could be greatly improved, leading to a more comprehensive understanding of groundwater dynamics.

Response: Thank you for your valuable comments on our study. We have carefully reviewed your suggestions and made corresponding revisions, and we hope these modifications meet your expectations. We agree that key factors such as hydrogeological conditions and environmental variables may significantly influence the model outputs and their interpretation. However, the core focus of this study is on building LSTM models for each monitoring site individually, aiming to simulate the temporal variation of groundwater level at the point scale. Within this framework, spatially fixed attributes such as aquifer properties and topography remain relatively stable over the short term and are unlikely to exert dynamic influence on the time series at a single site. Additionally, factors such as vertical leakage and surface water interactions are difficult to quantify due to limited data availability. In future work, if data conditions permit, we will consider incorporating these variables to enhance the physical interpretability and predictive accuracy of the model.

Specific comments:

Line 25: Please define NSE upon first mention to ensure clarity for readers unfamiliar with the metric.

Response: Thank you for pointing out this issue. We have added the full term "Nash-Sutcliffe Efficiency" when "NSE" first appears in the abstract.

Line 39: Provide more detailed justification of why monitoring groundwater levels is crucial, not only for managing water resources but also for protecting ecological

systems. Additionally, consider using the ML-predicted results to present a case study with quantitative analysis to better illustrate the implications.

Response: Thank you for the valuable comments from the reviewer. Following the suggestions, we have comprehensively revised the relevant parts of the manuscript to further elaborate on the importance of groundwater level depth, especially emphasizing its role in water resource management and ecosystem protection. Additionally, we supplemented the citations with the study by Liu et al. (2022), which used machine learning to predict groundwater level depth in the lower Tarim River, providing a quantitative case validation of the practical significance of groundwater level prediction. The revised content is located from line 46 to line 61.

Lines 62–67: The key disadvantage of physical models, compared to ML models, lies in their time-consuming setup, calibration, and validation processes. However, physical models have the advantage of offering more mechanistic insight into underlying hydrological processes, which ML models often lack.

Response: We thank the reviewer for highlighting the insufficient discussion on the comparison between physical models and machine learning models in the current manuscript. In response to your suggestion, we have revised and supplemented the relevant content accordingly. In the updated version, we have clearly stated the advantages of physical models in revealing the physical mechanisms of hydrological processes, while also acknowledging their limitations in regions with complex geological conditions due to high modeling complexity and substantial data requirements. The revised content is located from line 74 to line 84.

Line 118: The model would benefit from incorporating a wider range of influencing factors, such as aquifer properties, topography, hydraulic conditions (e.g., lateral flow, vertical leakage, groundwater storage, surface water interactions), and anthropogenic variables like population density. Spatial heterogeneity in evapotranspiration and precipitation should also be considered to improve model realism.

Response: We sincerely thank the reviewer for the professional and insightful comments. We fully agree that a variety of natural and anthropogenic factors—such as aquifer properties, topography, groundwater dynamics, population density, and evapotranspiration—can exert significant influence on regional groundwater level changes. However, considering the design rationale and actual data availability in this study, we have carefully reflected on and responded to this point from the following two perspectives:

First, the core framework of our study is to independently construct an LSTM model for each monitoring well to simulate the temporal variation of groundwater level at that specific location. The model uses historical meteorological variables and anthropogenic dynamic factors (including air temperature, precipitation, snow depth, and groundwater extraction) as inputs, aiming to capture the nonlinear response relationship between these temporally dynamic factors and groundwater level changes. Under this modeling strategy, spatially static attributes such as aquifer properties and topography remain constant over short periods at a given site and thus cannot provide dynamic explanatory power for the temporal evolution of groundwater levels at that point. Additionally, the spatial heterogeneity of factors such as evapotranspiration and precipitation primarily influences regional-scale patterns or spatial distributions. Since our study focuses on site-specific time series modeling and identification of dominant influencing factors, it is relatively less dependent on spatially heterogeneous variables. We have clarified this limitation in Section 3.5 "Model Limitations" of the revised manuscript.

Second, regarding the absence of variables related to groundwater dynamics (e.g., lateral flow, vertical leakage, and surface–groundwater interactions), we fully acknowledge their critical roles in groundwater system evolution. Although in theory, groundwater flow fields could be constructed through spatial interpolation of observed water levels, in our study the groundwater level is the target output variable of the model. Thus, prior to obtaining the model predictions, it cannot serve as an input driver.

Moreover, in practice, there is a lack of independent observational data (such as hydraulic gradients or recharge–discharge rates) that directly reflect groundwater dynamics, making it currently unfeasible to incorporate these factors into the model. In future work, if data availability improves, we intend to include such variables as key supplementary inputs to enhance the model's physical interpretability.

Figure 2: Consider including a geological map that shows the distribution of geological formations or aquifer types. This would help contextualize the results spatially.

Response: We thank the reviewer for the suggestion. In response, we have added a new subfigure to Figure 2 showing the distribution of the aquifer system. The corresponding description of the aquifer system distribution has been added in the revised manuscript (Lines 168–172). The revised figure is as follows:

[Figure]

Spatial distribution of the ground surface elevation (a), topography (b) and aquifer system (c) in the Songnen Plain, China.

Figure 4: The observed and simulated groundwater levels do not align well; the simulated series appears overly variable. Please explain the possible causes of this discrepancy, such as overfitting, lack of key input variables, or limitations in the model's temporal resolution.

Response: We thank the reviewer for the valuable comments. Although the original manuscript included an explanation for the poor model performance at certain monitoring wells, the reasoning lacked clarity and failed to accurately convey the sources of model error. In response, we have revised and reorganized the relevant paragraph to enhance its logical structure and coherence. The revised content is located from line 341 to line 363.

Lines 373–376 and 557–558: These sections are overly descriptive. Instead of simply stating observations, clarify what the results reveal about the status or trends of water resources. Quantitative insights or implications for water management should be emphasized.

Response: We sincerely thank the reviewer for the valuable comments. Based on the suggestions, we conducted an in-depth analysis of the implications of groundwater level dynamic classification for regional water resources management and supplemented the relevant content in Section 4.1 of the Discussion (lines 697–730). For different groundwater dynamic types, we explicitly proposed differentiated water resources management strategies. In addition, by comparing and linking our findings with previous related studies, we strengthened the scientific basis of the conclusions and demonstrated the consistency between the dynamic types identified in this study and existing empirical research results.

**References:**

Liu, Q., Gui, D., Zhang, L., et al.: Simulation of regional groundwater levels in arid regions using interpretable machine learning models, Sci. Total Environ., 831, 154902, https://doi.org/10.1016/j.scitotenv.2022.154902, 2022.